# The Bloom syndrome complex senses RPA-coated single-stranded DNA to restart stalled replication forks

Ann-Marie K. Shorrocks [1,2,6], Samuel E. Jones [1,2,6], Kaima Tsukada [1,2,3,6], Carl A. Morrow[1,2,6], Zoulikha Belblidia[1,2], Johanna Shen[1,2,4], Iolanda Vendrell[2,5], Roman Fischer [5], Benedikt M. Kessler [5] & Andrew N. Blackford [1,2✉]

The Bloom syndrome helicase BLM interacts with topoisomerase IIIα (TOP3A), RMI1 and RMI2 to form the BTR complex, which dissolves double Holliday junctions to produce non-crossover homologous recombination (HR) products. BLM also promotes DNA-end resection, restart of stalled replication forks, and processing of ultra-fine DNA bridges in mitosis. How these activities of the BTR complex are regulated in cells is still unclear. Here, we identify multiple conserved motifs within the BTR complex that interact cooperatively with the single-stranded DNA (ssDNA)-binding protein RPA. Furthermore, we demonstrate that RPA-binding is required for stable BLM recruitment to sites of DNA replication stress and for fork restart, but not for its roles in HR or mitosis. Our findings suggest a model in which the BTR complex contains the intrinsic ability to sense levels of RPA-ssDNA at replication forks, which controls BLM recruitment and activation in response to replication stress.

[1] MRC Weatherall Institute of Molecular Medicine, University of Oxford, John Radcliffe Hospital, Oxford OX3 9DS, UK. [2] MRC Oxford Institute for Radiation Oncology, Department of Oncology, University of Oxford, Oxford OX3 7DQ, UK. [3] Department of Transdisciplinary Science and Engineering, School of Environment and Society, Tokyo Institute of Technology, Tokyo 152-8550, Japan. [4] Department of Molecular, Cellular and Developmental Biology, Yale University, New Haven, CT 06520, USA. [5] Target Discovery Institute, Centre for Medicines Discovery, Nuffield Department of Medicine, University of Oxford, Oxford OX3 7FZ, UK. [6] These authors contributed equally: Ann-Marie K. Shorrocks, Samuel E. Jones, Kaima Tsukada, Carl A. Morrow. ✉email: andrew.blackford@imm.ox.ac.uk

Bloom syndrome is a rare hereditary chromosomal instability disorder caused by mutations in the gene encoding the BLM helicase[1]. It is characterized by growth retardation, immunodeficiency, hypersensitivity to sunlight, and high cancer predisposition[2]. Remarkably, Bloom syndrome patients are predisposed to the full spectrum of malignancies found in the general population, in contrast to other inherited cancer-predisposing disorders[3]. Cells from these patients display multiple signatures of genome instability, in particular a large increase in sister chromatid exchanges (SCEs)[4], which consequently serves as a diagnostic test for Bloom syndrome[5].

BLM functions in cells with TOP3A, RMI1, and RMI2, which together form the BTR "dissolvasome" complex that can process HR intermediates to prevent genetic crossovers[6]. Accordingly, mutations in TOP3A, RMI1, and RMI2 have been identified in patients with conditions similar to Bloom syndrome[7,8]. BLM has been proposed to act as a tumor suppressor by preventing crossovers between homologous chromosomes, which could lead to loss-of-heterozygosity[9,10]. However, more recent data from genome-wide analyses suggest that loss-of-heterozygosity events in BLM-deficient cells are extremely rare[11], indicating that other cellular functions of BLM may contribute to the increased cancer predisposition in Bloom syndrome patients. Such functions include an early-stage role in the HR repair process by stimulating DNA-end resection[12], processing of ultra-fine DNA bridges (UFBs) between sister chromatids in anaphase[13], and promoting restart of stalled replication forks[14]. Currently, no separation-of-function mutations have been identified that could distinguish the roles of BLM in double Holliday junction (dHJ) dissolution, DNA-end resection, UFB processing, and stalled fork restart, to prove which contribute to tumor suppression in a suitable animal model. Furthermore, very little is understood as to how the activities of the BTR complex are regulated in response to genotoxic stress, and how BLM can tell the difference between normal DNA metabolic processes and DNA damage or stalled replication forks.

RPA is an abundant, high-affinity ssDNA-binding complex that is thought to recognize ssDNA rapidly in cells whether it is generated during DNA replication or as a result of DNA damage[15]. It consists of three subunits, RPA1, RPA2, and RPA3 (also referred to as RPA70, RPA32 and RPA14 respectively, based on how they migrate during sodium dodecyl sulfate-polyacrylamide gel electrophoresis (SDS-PAGE)). All three subunits contain oligonucleotide (OB)-binding folds that either bind to ssDNA or mediate protein–protein interactions, with the N-terminal OB-fold of RPA1 and the C-terminal winged-helix domain of RPA2 being the major interaction sites recognized by proteins that respond to DNA damage or replication stress.

BLM complexes isolated from cell extracts contain RPA[16], and in early studies the N-terminal 447 residues of BLM were suggested to bind directly to the RPA1 subunit[17,18]. However, later work suggested that RMI1 rather than BLM mediates RPA association with the BTR complex[19]. Thus, the RPA-binding sites within the BTR complex have not yet been precisely mapped and the physiological relevance of its association with RPA has not yet been defined.

Here, we demonstrate that the Bloom syndrome complex contains three RPA-binding motifs, two in BLM and one in RMI1. We show that all three motifs contribute to RPA-binding of the BTR complex, and that they all interact with the same N-terminal domain of RPA1, indicating that the BTR complex can associate with at least three discrete RPA complexes. Using a combination of CRISPR-Cas9, RNA interference, and ectopic expression of recombinant proteins delivered by lentivirus, we generated a system to produce cells expressing a BTR complex that cannot bind RPA. Characterization of these cells revealed that RPA-binding is required for stable recruitment of BLM to DNA damage sites, stalled replication fork restart, and cellular resistance to replication stress, but not for the roles of BLM in HR or UFB processing.

## Results

**Identification of conserved RPA-binding motifs in the BTR complex.** The biochemical activities of the BTR dissolvasome complex have been extensively characterized in earlier studies (reviewed in ref. [6,20]). However, very little is known about how the functions of this protein complex are regulated in cells. To address this question, we focused our attention on the intrinsically disordered regions of the BTR complex, the functions of which are still poorly understood. Regions of intrinsic structural disorder are hotspots for post-translational modifications and often contain short evolutionarily conserved linear peptide motifs that can act as binding sites for other proteins[21].

To identify such motifs in the BTR complex, we performed amino-acid sequence alignments of all four BTR subunits (BLM, TOP3A, RMI1, and RMI2) from representative vertebrate orthologues (Supplementary Fig. 1a–d). These revealed the presence of six peptide motifs in the complex that have been highly conserved throughout evolution (four in BLM and two in RMI1; Fig. 1a and Supplementary Fig. 1a, b). We did not find any putative peptide motifs in TOP3A as this protein appears to consist only of its topoisomerase domain and a series of structured zinc-finger motifs[22]; nor did we identify peptide motifs in RMI2, as this protein is relatively small and made up of a single oligonucleotide-binding (OB)-fold domain[23,24] (Supplementary Fig. 1c, d).

We hypothesized that the conserved peptide motifs we identified in BLM and RMI1 mediate interactions between the BTR complex and other proteins. In support of this idea, motif 2 has previously been shown to bind to the DNA damage response mediator protein TOPBP1 when phosphorylated on Ser304[25,26]. We therefore designed synthetic biotinylated peptides corresponding to residues in all six motifs for use as baits in pulldown experiments. In addition, we examined these motifs for the presence of confirmed phosphorylation sites (defined as having been identified in at least two separate studies collated at https://www.phosphosite.org), but only motif 2 contained such a modification in a conserved residue (Ser304, as mentioned above). We therefore synthesized both native and phospho-Ser304 versions of motif 2 for use in subsequent experiments.

Biotinylated peptides were conjugated to streptavidin-coupled beads and incubated with HeLa nuclear extracts, and potential interacting partners were identified by liquid chromatography-tandem mass spectrometry (LC-MS/MS; Supplementary Data 1–7). Importantly, TOPBP1 was identified as one of the top hits in the pulldown with motif 2, but only when the peptide derived from this motif was phosphorylated on Ser304 (Supplementary Data 2, 3). In contrast, peptides corresponding to motifs 4 and 6 did not interact with any proteins other than common contaminants[27] (Supplementary Data 5, 7). Strikingly, the top hits from the pulldowns with peptides based on motifs 1 and 3 (from BLM), and motif 5 (from RMI1), were all three subunits of RPA (Fig. 1b and Supplementary Data 1, 4, and 6). We confirmed these results by immunoblotting (Fig. 1c), leading us to speculate that the BTR complex contains at least three separate interaction sites for RPA: two in BLM, and one in RMI1.

To test this, we investigated whether mutation of these motifs in the context of full-length BLM and RMI1 affected their interaction with RPA. First, 293FT cells were transfected with plasmids expressing GFP-tagged wild-type (WT) RMI1 or mutant RMI1 in which motif 5 was deleted. GFP immunoprecipitations (IPs) from lysates from these cells revealed that the mutant RMI1

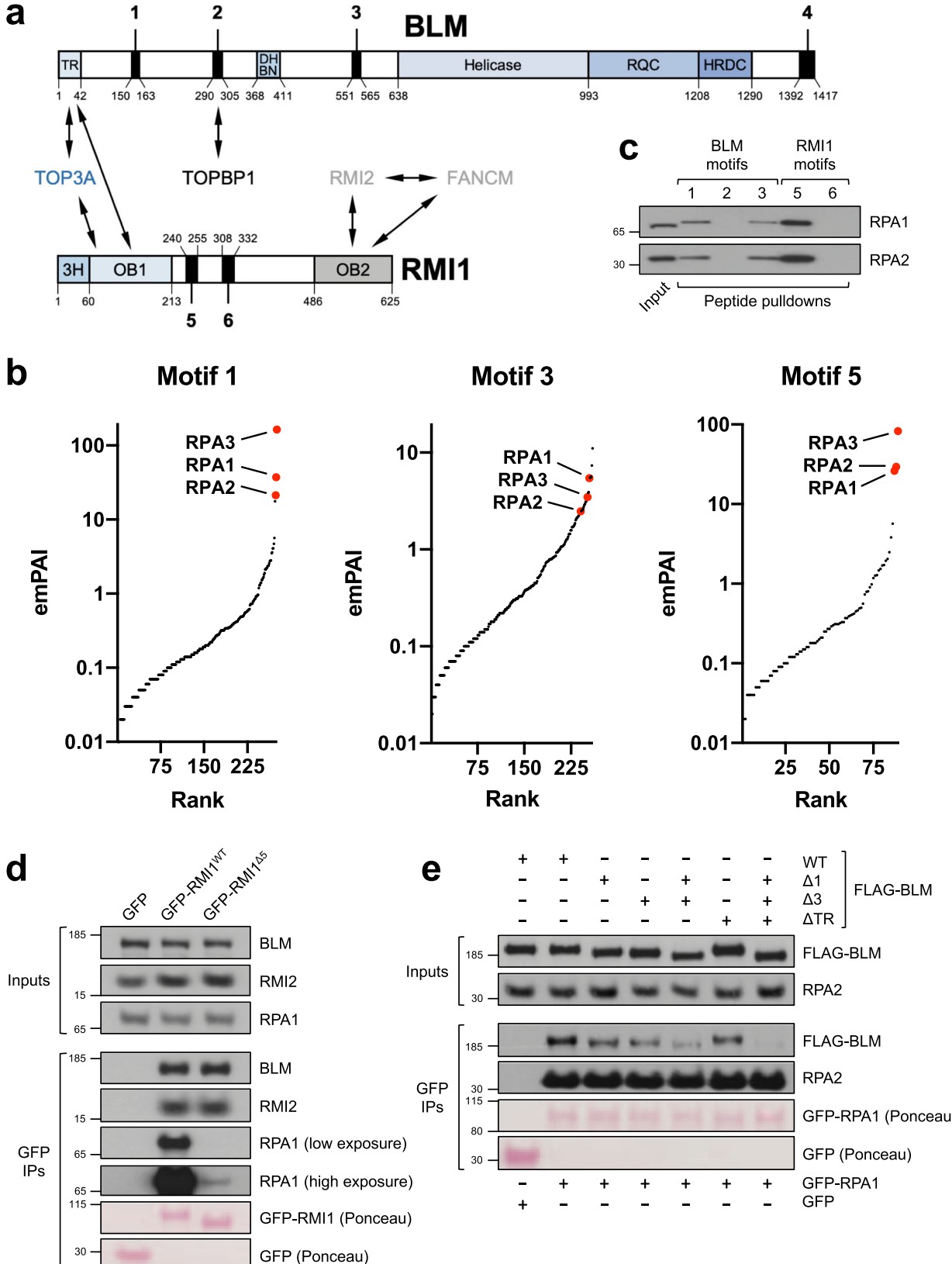

protein was severely deficient in its ability to interact with RPA (Fig. 1d). Conversely, its interactions with BLM and RMI2, which bind to other regions of RMI1[28,29], were unaffected, indicating that loss of motif 5 does not cause general misfolding of RMI1. Notably, at higher exposures of the RPA1 blot in this experiment, it was clear that some residual RPA still associated with the

mutant GFP-RMI1 protein above the level observed in the GFP-only lane, probably indirectly via RPA-binding motifs in BLM (see below).

Next, we deleted motifs 1 and 3 alone or in combination in constructs expressing FLAG-tagged BLM, transfected them into 293FT cells along with GFP or GFP-tagged RPA1, and performed

**Fig. 1 Identification of conserved RPA-binding motifs in the BTR complex. a** Schematic of proteins, domains, and motifs in the BTR complex. *TR* TOP3A-RMI1-binding domain, *DHBN* dimerization helical bundle in N-terminal, *RQC* RecQ C-terminal, *HRDC* helicase and RNaseD C-terminal; *3H* 3-helix bundle, *OB* oligonucleotide-binding. **b** Plots of MS hits from peptide pulldowns with motifs 1, 3, and 5, ranked by emPAI score[116] (number of unique peptide sequences adjusted for protein size). RPA subunits are highlighted in red. **c** Validation of MS results by western blotting, showing that RPA from HeLa nuclear extracts is specifically pulled down by biotinylated peptides based on motifs 1, 3, and 5 but not motifs 2 and 6. **d** GFP-pulldowns from 293FT cells transfected with constructs expressing GFP or the indicated GFP-tagged RMI1 variants, showing that loss of motif 5 disrupts RPA-binding. **e** GFP-pulldowns from 293FT cells transfected with constructs expressing either GFP or GFP-RPA1, and the indicated FLAG-tagged BLM variants, showing how loss of motifs 1 and/or 3, and the N-terminal TOP3A-RMI1 (TR) binding domain of BLM impacts on RPA-binding.

GFP IPs from cell lysates. Interestingly, loss of either motif 1 or motif 3 caused a marginal reduction in binding of BLM to RPA, whereas deletion of both BLM motifs caused a far more pronounced defect in RPA binding (Fig. 1e).

Given that some residual BLM-RPA binding was still apparent even in absence of both motifs 1 and 3, we considered the possibility that the presence of endogenous RMI1 (with its motif 5 that also binds RPA) in the BTR complex might be responsible for this. We therefore needed to design a BLM construct missing key residues in its TOP3A/RMI1 association domain. It has previously been shown that the first 133 residues of BLM are required for it to interact with the rest of the BTR complex[25,30]. However, motif 1 is located in close proximity to this region, raising the possibility that deletion of BLM residues 1–133 might interfere with RPA-binding regardless of BLM interaction with RMI1. To narrow down the binding site for BTR components on BLM, we examined the human BLM sequence for the presence of conserved residues known to disrupt binding of Sgs1, the budding yeast orthologue of BLM, to Top3 and Rmi1[31] (Supplementary Fig. 2a). This led us to predict that deletion of a region spanning residues 4–37 in the TOP3A/RMI1-binding ("TR") domain of BLM would disrupt its association with BTR complex components, and indeed this was the case (Supplementary Fig. 2b). We next examined the effect of this mutation (ΔTR) on BLM-RPA binding, in constructs either with or without motifs 1 and 3. Mutation of the TR domain of BLM caused a small decrease in RPA binding, indicating that stable BLM association with RPA depends to some extent on the presence of the RPA-binding motif in RMI1 (Fig. 1e). In support of this, mutation of the TR domain in combination with loss of motifs 1 and 3 abrogated almost all RPA binding to BLM. Taken together, these results demonstrate that all three RPA-binding motifs in the BTR complex contribute to maximize the association of BLM with RPA.

**The RPA-binding motifs in BLM and RMI1 interact with the RPA1 N-terminal OB-fold.** Given the presence of three RPA-binding motifs in BLM and RMI1, we wondered which subunit(s) of RPA these motifs interact with. To answer this question, we carried out further pulldown experiments with peptides derived from motifs 1, 3, and 5 in which one aromatic residue was replaced with the synthetic photoreactive amino-acid *p*-benzoyl-L-phenylalanine (BPA). UV-crosslinking and subsequent denaturing washes revealed that only the RPA1 subunit could be retained by these peptides (Fig. 2a), indicating that all three motifs interact directly with RPA1, but not RPA2 or RPA3.

The N-terminus of RPA1 contains an OB-fold domain with a basic binding cleft that mediates interactions with multiple DNA damage checkpoint and repair factors including p53, MRE11, ATRIP, RAD9, DNA2, ETAA1, and PRIMPOL[32–39]. Interestingly, when we compared the sequences of BLM/RMI1 motifs 1, 3, and 5 with the motifs from these proteins, we noticed significant sequence similarity (Fig. 2b). We thus speculated that BLM and RMI1 might also interact with the same site on RPA1. To test this, we in vitro translated the RPA1-RPA2-RPA3 complex with either WT or a mutant version of RPA1 in which key residues in the N-terminal basic binding cleft of RPA1 were

mutated (R41E/Y42F)[40]. We then incubated the protein products with biotinylated peptides derived from the three BLM/RMI1 motifs for subsequent peptide pulldowns. Results from these experiments demonstrated that while all three peptides could bind WT RPA, the R41E/Y42F mutation in RPA1 abolished these interactions (Fig. 2c).

To verify the importance of the N-terminal OB-fold domain of RPA1 in the context of full-length proteins in cells, we carried out pulldowns from 293FT cells transfected with constructs expressing GFP-tagged WT or R41E/Y42F RPA1. Results from this experiment showed that mutation of the RPA1 N-terminal OB-fold inhibited MRE11 binding as expected from previous work[33], and led to a complete loss of detectable binding of RPA1 to the BTR complex (Fig. 2d). Importantly, the RPA1-R41E/Y42F mutant was still proficient in binding to XPA, which associates with RPA via the central region of RPA1 and the C-terminus of RPA2[41,42].

We conclude from these data and those above that the BTR complex contains three acidic peptide motifs (two in BLM and one in RMI1) that interact with the same basic binding cleft in the N-terminal OB-fold of RPA1, raising the possibility that at least three separate RPA complexes can associate simultaneously with the BTR complex.

**Generation of a cell-based system to test the physiological relevance of BTR-RPA binding.** To investigate whether RPA-binding contributes to some or all of the functions of the BTR complex, we needed to generate cells expressing mutants of BLM and RMI1 in which all three RPA-binding motifs were deleted. To do this, we first generated *BLM* knockout RPE-1 cells using CRISPR-Cas9 (Fig. 3a). We also tried to obtain *RMI1* single and *BLM/RMI1* double knockout cells but these attempts were unsuccessful, in line with previous studies indicating that complete loss of RMI1 or its stable binding partner TOP3A is lethal in vertebrate cells, regardless of the presence of p53[43–46].

We selected three independent $BLM^{-/-}$ RPE-1 clones in which no BLM protein could be detected (Fig. 3a), for further characterization. Sequencing of the targeted alleles in all three clones confirmed unique indels resulting in frameshift mutations and premature stop codons in the *BLM* locus (Supplementary Data 8). BLM-deficient cells typically display a large increase in SCE levels, so we first analyzed these in our CRISPR-generated cell lines. We found that $BLM^{-/-}$ RPE-1 cells had a >10-fold increase in SCEs (Supplementary Fig. 3a), similar to Bloom syndrome patient cells[4], $Blm^{-/-}$ mouse embryonic fibroblasts[47] and $BLM^{-/-}$ chicken DT40 cells[48], thus confirming that these RPE-1 clones are true BLM knockout cells.

We also wished to know how our $BLM^{-/-}$ RPE-1 cells behaved in assays measuring other reported roles for BLM, in order to reveal the cellular role of RPA-binding in BTR complex functions. BLM has a key role in processing UFBs, and its absence leads to an increase in these structures in anaphase cells[13]. We therefore analyzed UFB numbers in WT and $BLM^{-/-}$ clones and found that as expected, our $BLM^{-/-}$ knockouts displayed around a threefold increase in UFBs (Supplementary Fig. 3b).

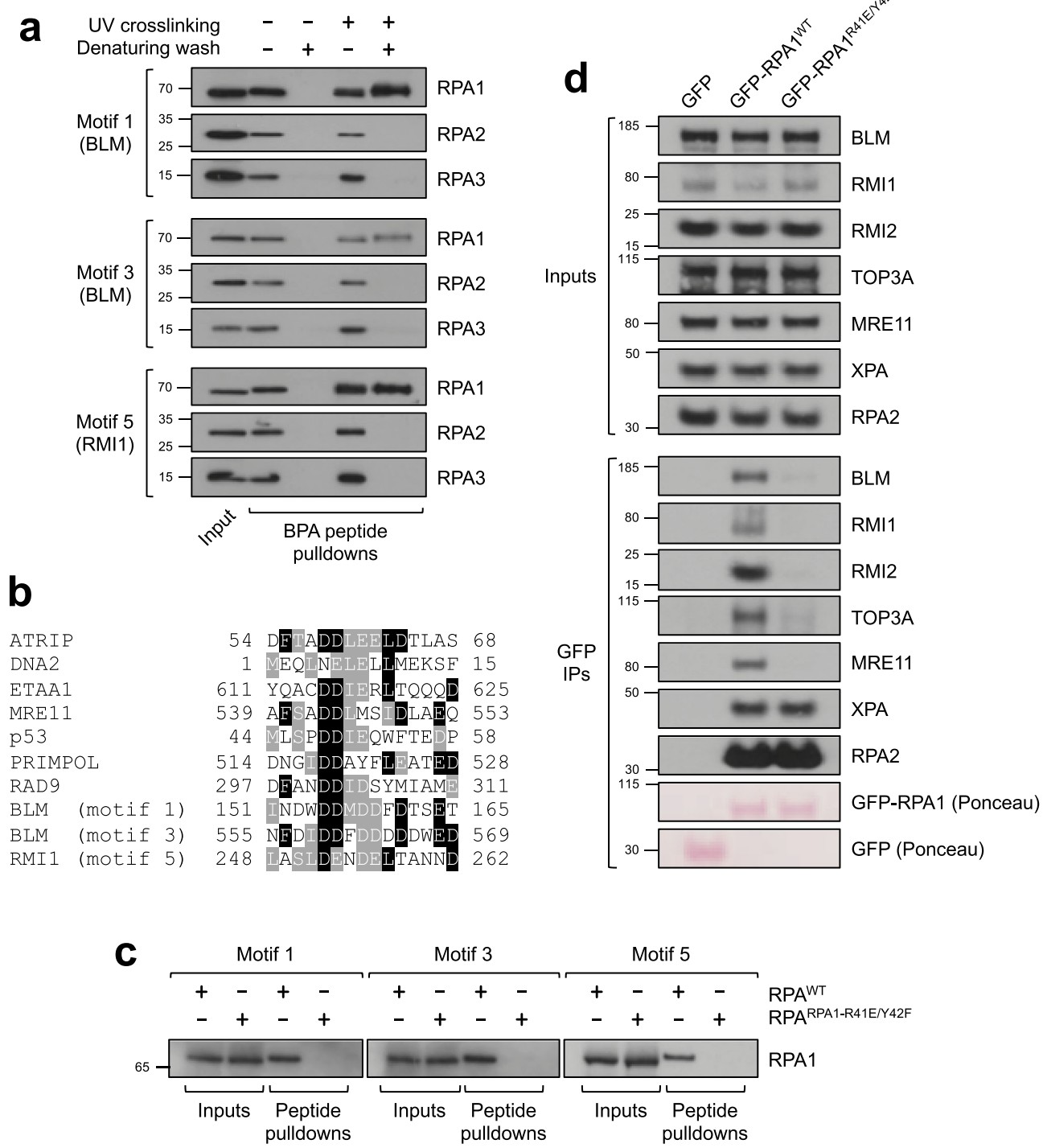

**Fig. 2 The RPA-binding motifs in BLM and RMI1 interact with the RPA1 N-terminal OB-fold. a** BPA-modified peptide pulldowns from HeLa nuclear extracts with UV-crosslinking and denaturing washes where indicated, demonstrating that motifs 1, 3, and 5 interact directly with the RPA1 subunit of RPA. **b** Sequence alignments showing the similarity of RPA1-binding motifs in DNA damage response proteins. **c** Peptide pulldowns from rabbit reticulocyte lysates containing in vitro translated heterotrimeric RPA complexes incorporating either WT or R41E/Y42F RPA1, showing that all three BLM/RMI1 motifs cannot interact with the mutant RPA complex. **d** GFP-pulldowns from 293FT cells transfected with constructs expressing GFP or the indicated GFP-tagged RPA1 variants, showing that mutation of the N-terminal OB-fold of RPA1 (R41E/Y42F) disrupts binding to the BTR complex and MRE11. XPA is a negative control because it binds to RPA via different domains.

BLM has also been implicated in promoting DNA-end resection in some studies[12,49–51], but not in others[52,53]. To test how BLM loss affects resection in RPE-1 cells, we used a fluorescence-activated cell-sorting (FACS)-based assay to measure RPA recruitment to chromatin in response to camptothecin (CPT) treatment in $BLM^{+/+}$ and $BLM^{−/−}$ cells[54]. Results from this assay demonstrated that $BLM^{−/−}$ cells are significantly impaired in DNA-end resection, but not by >~50% (Supplementary Fig. 3c). Thus, in our hands, BLM does promote resection but is not essential for this process. This is in line with previous

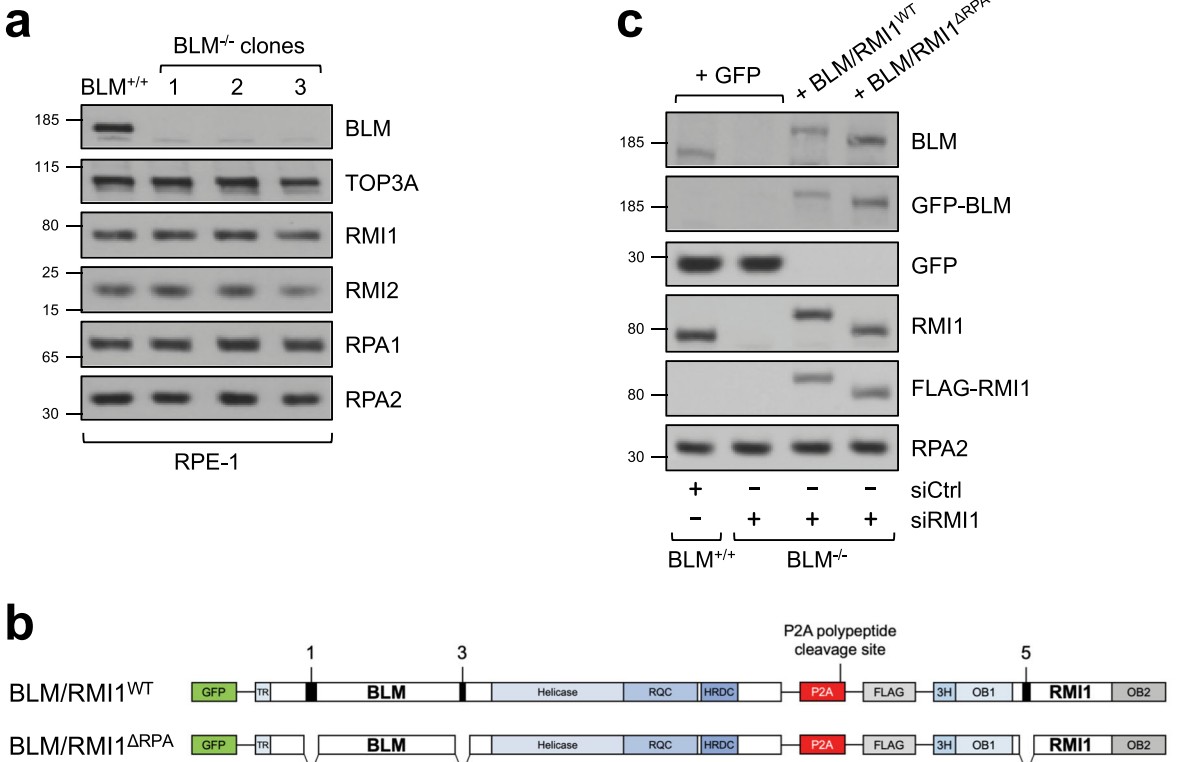

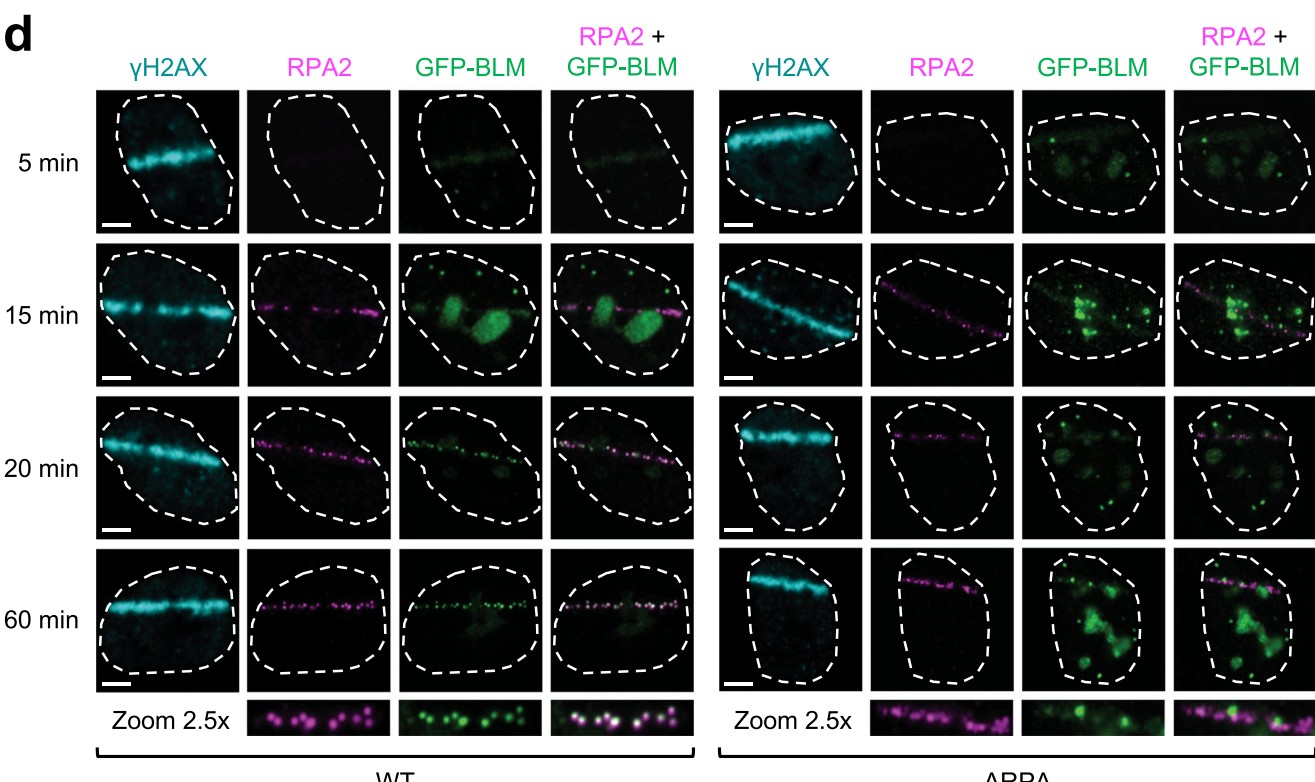

**Fig. 3 Generation and complementation of *BLM*$^{-/-}$ RPE-1 cells with RPA-binding mutants of BLM and RMI1. a** Western blots showing that BLM expression is undetectable in three separate *BLM* knockout RPE-1 clones. **b** Schematic showing the complementation strategy for adding back WT BLM/RMI1 proteins or mutant ones lacking RPA-binding motifs to *BLM*$^{-/-}$ cells treated with siRMI1. **c** Western blots showing the levels of exogenous GFP and BLM/RMI1 proteins delivered by lentivirus in WT or *BLM*$^{-/-}$ cells treated with siCtrl or siRMI1 as indicated. **d** Airyscan high-resolution confocal microscopy image time-course, showing the kinetics of recruitment of γH2AX, RPA2, and WT/mutant GFP-BLM proteins to laser-induced DSBs in representative cells. Zoomed-in images of laser lines are from the 60-min time-point images. Scale bars, 5 μm.

studies suggesting that BLM is functionally redundant with other RecQ helicases such as WRN and a parallel long-range resection pathway involving EXO1[12,50].

The role of BLM in maintenance of genome stability may be most important during S phase. Some BLM-deficient cell lines have been reported to be hypersensitive to genotoxic agents such as hydroxyurea (HU) that perturb DNA replication[55–58], although not all studies have found this[59,60]. We therefore investigated the requirement for BLM in response to DNA replication stress in our knockout cells, and found that all three of our $BLM^{-/-}$ RPE-1 clones were hypersensitive to HU treatment (Supplementary Fig. 3d). Using DNA fiber assays, BLM-deficient cells have been shown to be defective in replication fork restart after exposure to HU and other agents that interfere with DNA replication[14,61]. We confirmed that $BLM^{-/-}$ RPE-1 cells have a replication restart defect after HU treatment using similar analyses (Supplementary Fig. 3e, f). The fork restart defect was not due to increased RAD51 accumulation at stalled forks (as observed in budding yeast cells deficient in Sgs1[62]), because RAD51 foci formation after exposure to HU was no different regardless of the presence of BLM (Supplementary Fig. 3g, h).

To establish a complementation system to test the cellular requirement for RPA association with the BTR complex in these assays, we designed a lentiviral vector to express GFP-tagged BLM and FLAG-tagged RMI1 from the same promoter, separated by a P2A self-cleaving peptide sequence to produce two separate polypeptides (Fig. 3b). Mutations were then introduced to delete the two RPA-binding motifs in BLM and the one in RMI1. $BLM^{-/-}$ cells were then infected with viruses to express either WT GFP-BLM and FLAG-RMI1, or RPA-binding mutant versions of these proteins (BLM/RMI1$^{\Delta RPA}$), or GFP alone as a control. By subsequently depleting RMI1 transiently using siRNA, we were able to establish cells that do not detectably express BLM or RMI1, and cells expressing only RPA-binding mutant versions of these proteins for functional assays (Fig. 3c).

**RPA-binding stabilizes BLM at ssDNA damage sites but is not essential for BLM recruitment to DSBs.** Given that RPA is a platform for recruitment of multiple DNA damage response proteins in response to genotoxic stress[15], we considered the possibility that RPA-binding promotes recruitment of BLM to DNA damage sites. We therefore used laser microirradiation combined with 5-bromo-2'-deoxyuridine (BrdU) presensitization to induce such lesions, and examined GFP-BLM$^{WT}$ and GFP-BLM$^{\Delta RPA}$ recruitment in a time-course experiment (Fig. 3d). Interestingly, both proteins accumulated rapidly to laser lines (within 5 min), although at relatively low levels and with a diffuse staining pattern. Such early recruitment to DSBs has been described previously for BLM[63], and requires its C-terminus rather than the N-terminal region containing the RMI1 and RPA-binding sites. Given also that RPA recruitment was not detectable until after 5 min, it made sense to us that there was no difference between WT and ΔRPA GFP-BLM recruitment at this early time point. By 15 min, RPA microfoci appeared along the path of the laser line in ~50% of irradiated cells, in keeping with previous reports showing that such RPA-coated ssDNA tracts are produced as a result of DNA-end resection in S and G2 cells only[64,65], with a ~700 s half-life of recruitment[66]. Interestingly, by 20 min WT GFP-BLM also formed bright microfoci along the path of the laser line, which colocalized with RPA. In contrast, the GFP-BLM$^{\Delta RPA}$ protein instead remained diffuse and at a low level at this time-point, resembling its early localization pattern. Indeed, even though at later times (>60 min) the mutant protein formed a small number of discernible foci along the path of the laser line, these rarely colocalized with RPA. Therefore, we

conclude that at least one major function of RPA-binding is to stably recruit the BTR complex to ssDNA damage sites in cells, although BLM can still localize to DSBs through other mechanisms.

**RPA-binding is required for the BTR complex to restart stalled replication forks but not to suppress SCEs, process UFBs, or promote DNA-end resection.** Given that BLM recruitment to RPA-ssDNA sites was defective in absence of the RPA-binding motifs in BLM and RMI1, we next examined whether any of the biological processes that depend on BLM was defective in cells expressing a BTR complex defective in RPA binding. We first examined the level of SCEs in cells expressing BLM and RMI1 that cannot bind RPA. Although $BLM^{-/-}$ cells depleted of RMI1 and complemented with GFP alone displayed the expected 10-fold excess of SCEs, cells complemented with either exogenous WT or the RPA-binding mutant BLM/RMI1 proteins had levels of SCEs similar to WT RPE-1 cells (Fig. 4a). Thus, RPA association with the BTR complex is not required for suppression of SCEs. This is in line with the fact that dHJs do not contain ssDNA and our observation that BLM can still localize to DSB sites in the absence of the RPA-binding motifs in BLM and RMI1, although at a reduced level (Fig. 3d).

We then checked whether BLM-RPA binding is required for UFB processing. RPA is not observed on all bridges, and although RPA recruitment to UFBs is BLM-dependent, BLM and RPA do not colocalize on these structures[67,68]. Furthermore, recent biochemical evidence suggests that RPA may actually exclude BLM once deposited on UFBs[69]. In line with these studies, in cells expressing RPA-binding BLM/RMI1 mutants, GFP-BLM was still able to localize to UFBs (Fig. 4b), and the number of UFBs was similar to that observed in cells expressing WT BLM (Fig. 4c), indicating that BLM-RPA binding is not required for the role of the BTR complex in UFB processing.

Next, we examined whether BTR-RPA binding is required for DNA-end resection. Similar to the situation with SCEs and UFBs, we found that the RPA-binding mutant BLM/RMI1 proteins were able to complement the mild resection defect of BLM/RMI1-deficient cells (Fig. 4d), indicating that RPA-binding is not required for the BTR complex to stimulate DNA-end resection. This is in line with the observation that a small amount of BLM localizes to DSBs prior to the appearance of RPA foci (Fig. 3d).

We then examined whether RPA-binding is required for the role of the BTR complex in promoting replication fork restart. Using DNA fiber analyses as described above (Supplementary Fig. 3e), we found that in contrast to WT proteins, which could complement BLM/RMI1-deficient cells, cells expressing the BLM/RMI1$^{\Delta RPA}$ mutants were as defective in fork restart and displayed a similar increase in stalled forks after HU as cells lacking BLM/RMI1 expression altogether (Fig. 5a and Supplementary Fig. 4). In line with this, the mutant proteins were also unable to complement the HU hypersensitivity of BLM/RMI1-deficient cells (Fig. 5b). Thus, we conclude that RPA-binding is specifically required for the role of the BTR complex in promoting replication fork restart, rather than its roles in suppression of SCEs, UFB processing, or DNA-end resection.

**RPA-binding is required for BLM recruitment to stalled replication forks.** Given that RPA-binding promotes stable BLM recruitment to ssDNA foci at DSB sites (Fig. 3d), we considered the possibility that RPA-binding might be important for recruitment of BLM to stalled replication forks in response to HU, as nucleotide depletion by this agent leads to helicase-polymerase uncoupling and accumulation of RPA-ssDNA[64]. To test this, we examined the colocalization of WT and RPA-binding

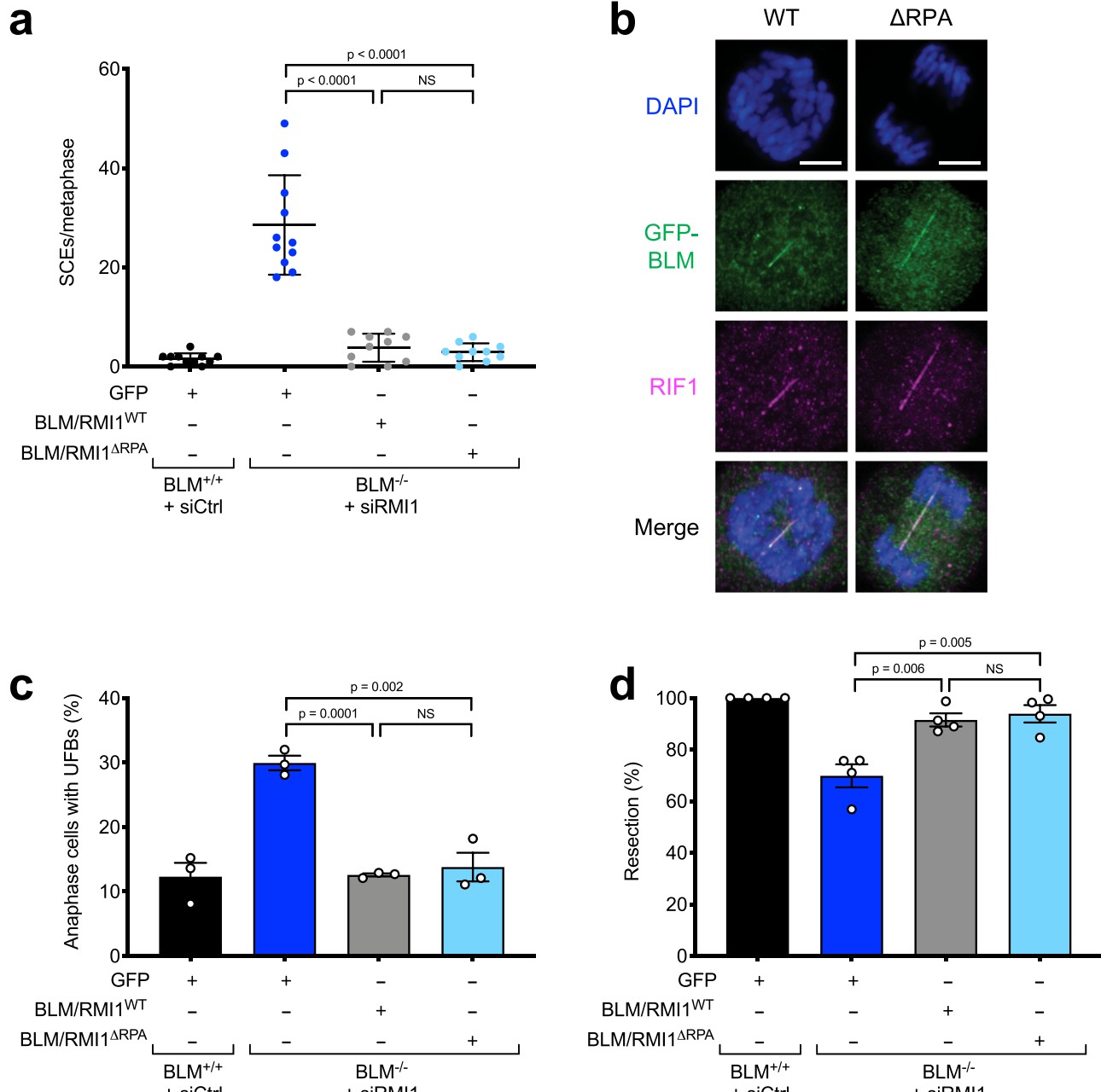

**Fig. 4 RPA-binding is not required for the BTR complex to suppress SCEs, process UFBs, or promote DNA-end resection. a** RPA-binding by BLM and RMI1 is not required for suppression of SCEs ($n = 3$, one representative experiment shown). At least 10 metaphases were scored per sample. Error bars denote standard deviation (SD) from the mean. Significance was determined using the two-sided Mann–Whitney $U$ test. **b** Airyscan high-resolution confocal microscopy images showing that both WT and RPA-binding (ΔRPA) mutant BLM proteins localize to UFBs (marked by RIF1, which localizes independently of BLM to UFBs[68]). Scale bars, 5 μm. **c** RPA-binding by BLM and RMI1 is not required for suppression of UFBs. Error bars denote standard error of the mean (SEM). Significance was determined by two-sided unpaired $t$ test ($n = 3$). **d** RPA-binding by BLM and RMI1 is not required for DNA-end resection. Error bars denote SEM. Significance was determined by two-sided unpaired $t$ test ($n = 4$).

mutant GFP-BLM proteins with RPA foci formed in response to HU treatment by immunofluorescence microscopy. Results from these analyses indicated that there was a significant defect in the ability of the GFP-BLM$^{ΔRPA}$ mutant protein to form foci at RPA-ssDNA sites in response to replication stress (Fig. 5c, d).

Thus, we conclude that RPA-binding promotes stable recruitment of BLM to stalled replication forks, where it is required for replication fork restart and genome stability after replication stress. Furthermore, given that the BTR complex contains three discrete binding sites for RPA1, our data suggest a model in which BTR contains the intrinsic ability to measure the level of RPA-ssDNA at

replication forks to control BLM recruitment and activation in response to replication stress (Fig. 6a, b).

## Discussion

In this report, we have precisely delineated how the Bloom syndrome complex interacts with RPA by carrying out unbiased proteomic analyses of conserved peptide motifs. We found that BLM contains two RPA-binding motifs, and RMI1 contains a third (Fig. 1b, c). Interestingly, these motifs all recognize the same N-terminal domain on the RPA1 subunit of the RPA

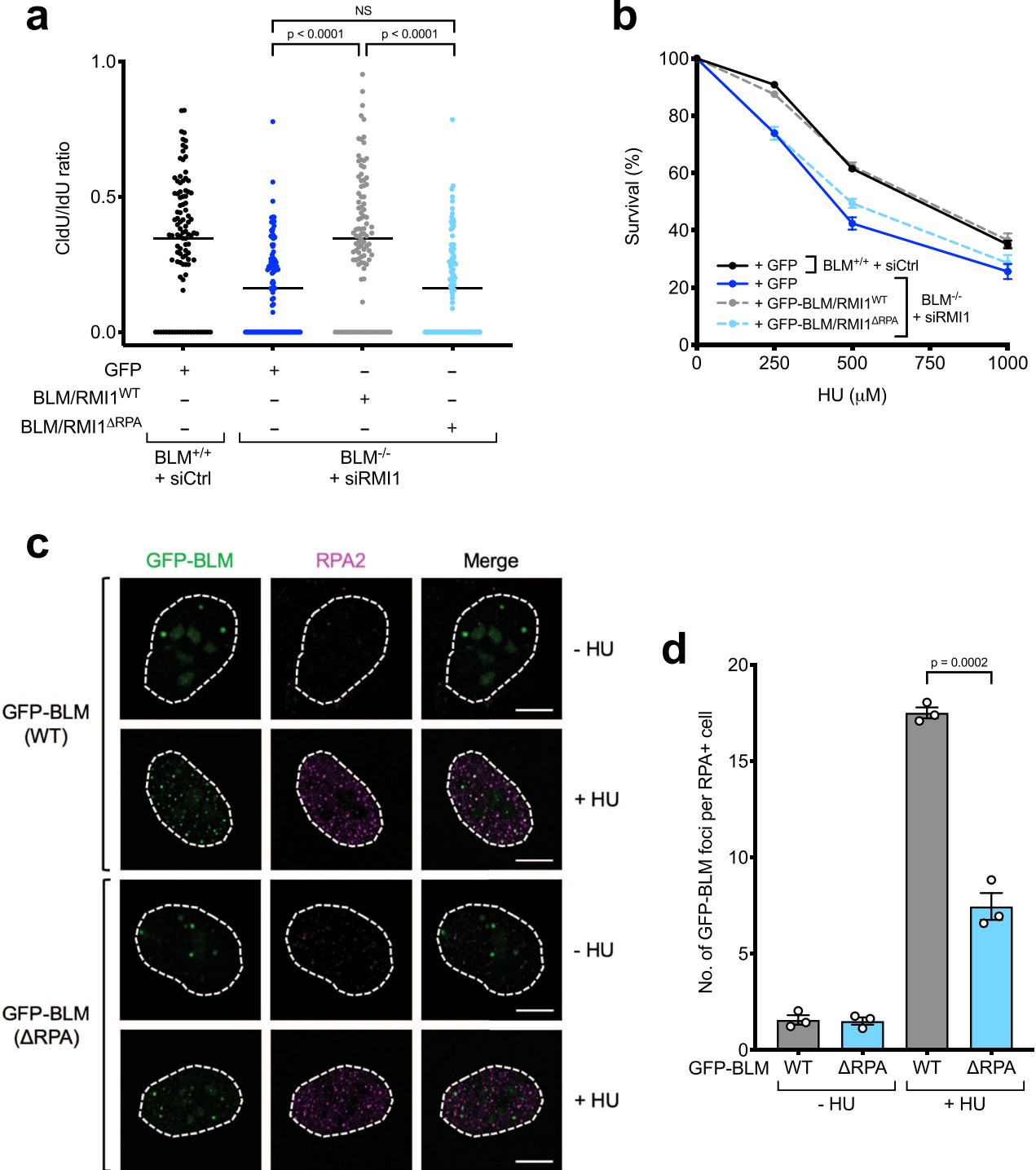

**Fig. 5 RPA-binding is required for the role of the BTR complex in response to DNA replication stress. a** Quantification of CldU/IdU tract length ratios in DNA fiber assays showing that RPA-binding by BLM and RMI1 is required for replication fork restart ($n = 3$, one representative experiment shown). At least 100 DNA fibers were analyzed per sample. Significance was determined using the two-sided Mann–Whitney $U$ test. **b** Colony survival assays showing RPA-binding by BLM and RMI1 is required for HU resistance ($n = 3$, error bars denote SEM). Cells were treated with the indicated doses of HU for 24 h. **c** Immunofluorescence confocal microscopy images of the indicated proteins, demonstrating that the RPA-binding BLM mutant (ΔRPA) is defective in its ability to form foci that colocalize with RPA2 in response to replication stress. Cells were treated where indicated with 2 mM HU for 24 h. **d** Quantification of the number of WT and ΔRPA GFP-BLM foci in the experiment shown in **c**). Significance was determined using the two-sided unpaired $t$ test ($n = 3$, error bars denote SEM).

heterotrimer (Fig. 2a–d). Crucially, all three motifs contribute to RPA-binding in the context of full-length BLM and RMI1 proteins (Fig. 1d, e), indicating that they are not functionally redundant. Instead, our data suggest that the BTR complex requires interaction with three RPA complexes for stable

recruitment to ssDNA sites via motifs in both BLM and RMI1 (Figs. 3d and 5c, d), which explains why BLM recruitment to DNA damage foci depends on RMI1 and vice versa[65,70]. Notably, given that it is still unclear what the stoichiometry is of the various subunits within the BTR complex (BLM for example can

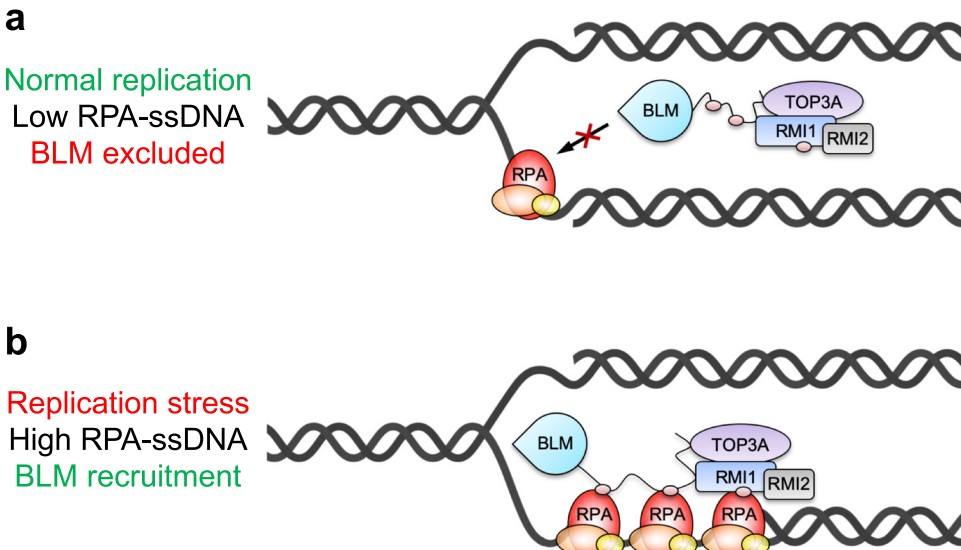

**Fig. 6 Model for the role of RPA-binding in specifically recruiting the BTR complex to stalled replication forks. a** During normal DNA replication, the BTR complex is not recruited to ongoing replication forks because not enough RPA-coated ssDNA complexes are present. **b** When a replication fork stalls, increased RPA-ssDNA leads to stable recruitment of the BTR complex and replication fork restart. Gray spheres represent the three RPA-binding motifs in BLM and RMI1.

dimerize via its DHBN domain[71]), the BTR complex in cells could potentially associate with more than three RPA complexes.

BLM was originally shown to interact with RPA using purified recombinant proteins in vitro[17]. Subsequently, it was demonstrated that RPA is present in BTR complexes immunoprecipitated from cells[16], and that the N-terminus of BLM is required for RPA-binding[18]. It was recently suggested that acidic peptides from BLM (similar to motifs 1 and 2 in our study; Fig. 1a) can interact with a recombinant N-terminal fragment of RPA1[72]. However, motif 2 is in fact the TOPBP1-binding site on BLM[25], and in line with this we did not detect any interaction of this motif with RPA (Fig. 1c), nor did deletion of this motif from the full-length BLM protein have any effect on RPA binding (Supplementary Fig. 5a). A different group could not detect any interaction between recombinant BLM and RPA, and instead mapped a binding site for RPA on RMI1[19]. Interestingly, that site corresponds to what we designated as motif 6 in our current study (Fig. 1a). However, in our hands the peptide corresponding to this motif does not interact with RPA (Fig. 1c); furthermore, deletion of this motif in full-length RMI1 does not impact on RPA binding, unlike deletion of motif 5, which almost abrogates the RMI1–RPA interaction (Fig. 1d and Supplementary Fig. 5b).

Some biochemical studies have suggested that RPA-binding stimulates the dHJ dissolution reaction by the BTR complex[19,73], whereas others have contradicted this[69,74]. We found that in human cells, RPA-binding by the BTR complex is probably not required for dHJ dissolution because cells expressing mutant BLM and RMI1 that cannot bind RPA do not display increased SCEs (Fig. 4a). This made sense to us, given that dHJ structures do not contain ssDNA and would therefore not be expected to require RPA-binding to recruit the BTR complex to dissolve them. Likewise, we found that direct binding of RPA to the BTR complex is not required for DNA-end resection (Fig. 4d). In vitro, BLM can promote resection by two nucleases, EXO1 and DNA2[75,76]. Genetically, however, loss of BLM is epistatic with DNA2 depletion and additive with EXO1 deficiency[12,50], indicating that in cells BLM cooperates with DNA2 to promote resection. This is in line with studies in budding yeast, where Sgs1 cooperates with Dna2, and Exo1 works in a parallel resection pathway[12,77,78]. Interestingly, an interaction between DNA2 and

RPA is conserved throughout evolution and stimulates DNA2 nuclease activity[35,79–83]. Thus, it may be that RPA-binding by DNA2 is sufficient to promote resection by the BLM/DNA2 resectosome in cells even if RPA-binding by the BTR complex is lost. Finally, although the BTR complex plays a key role in UFB processing in anaphase, we found that RPA-binding does not recruit the complex to UFBs (Fig. 4b), nor does it appear to play a role in UFB processing (Fig. 4c). This is in line with recent evidence suggesting that RPA actively excludes BLM from ssDNA in mitosis[69], and that BTR recruitment to UFBs instead depends on the DNA translocase PICH[84].

Whereas RPA-binding is not required for all cellular roles of the BTR complex, we found that it does promote replication fork restart and cellular resistance to HU (Fig. 5a, b). The precise mechanism as to why BLM is required for fork restart is still unclear, but might for example involve fork remodeling by promoting fork regression or unwinding of aberrant DNA structures such as G-quadruplexes[85,86], which could be stimulated by RPA-binding[18]. In addition, our cell-based experiments indicate that a major role of RPA-binding is to stably recruit the BTR complex to stalled replication forks (Fig. 5c, d). This finding also suggests a mechanism whereby cells can "measure" the amount of RPA-coated ssDNA present at a stalled replication fork and engage the BTR complex to rescue it, while preventing precocious recruitment and activation of BLM at ongoing replication forks where it is not required (Fig. 6a, b). Our data also suggest that BLM can still localize to DNA damage sites through other mechanisms, where it is able to promote DNA-end resection at early times upon DSB induction (prior to RPA accumulation), and dissolution of dHJs (that do not contain ssDNA) at later times. One mechanism may involve recruitment of the BTR complex via FANCM, which can cooperate with BLM to suppress SCE and error-prone long-tract gene conversion[87–89].

It is interesting to note that interaction between a RecQ helicase and a ssDNA-binding protein has been highly conserved throughout evolution, possibly in order to respond to stalled replication forks. For example, immunodepletion of RPA from *Xenopus* egg extracts prevents the association of BLM with chromatin[90]. Furthermore, Sgs1 has been shown to interact with RPA via an acidic region in the N-terminus[91], and the fission

yeast ortholog of BLM, Rqh1, also forms a complex with RPA and has been suggested to be recruited to stalled forks via this inter-action[92]. Even in bacteria, it has been shown that RecQ interacts with SSB, and that this interaction is required for localization of the helicase to the replication factory[93].

Our data suggest that the fork restart defect observed in absence of BLM is not caused by excessive accumulation of RAD51 (as seen in Sgs1 mutant budding yeast[62]), because we did not detect any difference in RAD51 foci between WT and $BLM^{-/-}$ cells (Supplementary Fig. 3g, h). This is in line with previous cell-based studies demonstrating that BLM deficiency does not result in increased RAD51 recruitment at stalled forks or DSBs induced by ionizing radiation, even though the loss of BLM in certain HR-deficient genetic backgrounds can restore RAD51 foci to some extent through an unknown mechanism[52,94]. Although BLM has been shown to disrupt RAD51-ssDNA filaments in vitro, unlike other known anti-recombinases it can only do so when RAD51 is in its recombination-inactive, ADP-bound form[95]. In contrast, Sgs1 can readily dismantle Rad51-ssDNA filaments in vitro via a mechanism that is not coupled to the Rad51 ATP hydrolysis cycle[96], a function shared with mammalian RECQL5[97]. This suggests that BLM may have lost the anti-recombinogenic function of Sgs1 in disassembling RAD51-ssDNA filaments at some point during evolution, while RECQL5 and possibly other RecQ helicases have maintained it. The BTR complex may nonetheless have one or more anti-recombinogenic functions downstream of RAD51-mediated strand invasion. Although BLM can dissolve D-loops in vitro, it can only do so when they are protein-free[98,99], a situation that is unlikely to occur in cells. In contrast however, purified TOP3A-RMI1-RMI2 can readily dissolve D-loops either in the presence or absence of RAD51, RAD54, and RPA[100], and Sgs1-Top3-Rmi1 disrupts D-loops in vivo[101]. Future studies will be required to establish whether the BTR complex similarly has a role in disrupting D-loops in human cells, or whether its role in the late stages of HR is limited to promoting dissolution of late recombination intermediates into non-crossovers (as suggested previously[102]), rather than actually suppressing recombination per se. If the BTR complex does disrupt D-loops in cells, it will be interesting to establish whether RPA-binding by BLM/RMI1 is required for this putative function.

Bloom syndrome is a heterogeneous condition with many associated pathologies. BLM mutations found in patients either result in loss of BLM protein or inhibit its helicase activity, which is required for all its known functions in human cells. It is thus unclear which cellular roles of BLM are required to prevent the various symptoms of the disease. We have found that loss of RPA association with the Bloom syndrome complex constitutes a clear separation-of-function mutation with regard to the cellular functions of BLM, given that RPA-binding is required for replication fork restart but not for other roles in DNA repair. Interestingly, mutation of Thr99 in the N-terminus of human BLM (a TQ site that is a preferred substrate for the apical DNA damage response kinases ATM and ATR[103]) has also been reported to cause replication fork restart defects but not increased SCEs[14]; however, as Thr99 is not conserved beyond primates (Supplementary Fig. 1a), it cannot easily be used as a separation-of-function mutation to investigate the role of BLM in vivo. In contrast, future studies in animal models in which the RPA-binding motifs in BLM and RMI1 are mutated may be infor-mative in elucidating which function(s) of BLM are required to prevent the various pathologies with Bloom syndrome, with possible future therapeutic consequences for patients.

## Methods

**Sequence alignments**. The following NCBI reference sequences were used to generate alignments in Supplementary Fig. 1a–d: BLM, NP_000048.1 (human),

NP_031576.4 (mouse), XP_003434427.1 (dog), NP_001007088.2 (chicken), NP_001079095.1 (frog), XP_017207539.1 (zebrafish); RMI1, NP_079221.2 (human), NP_083180.3 (mouse), XP_013972800.1 (dog), NP_001026783.1 (chicken), NP_001086875.1 (frog), NP_956474.1 (zebrafish); RMI2, NP_689521.1 (human), NP_001156404.1 (mouse), XP_852348.2 (dog), NP_001006174.1 (chicken), XP_018091789.1 (frog), XP_691674.2 (zebrafish); TOP3A, NP_004609.1 (human), NP_033436.1 (mouse), XP_546656.2 (dog), NP_001025807.1 (chicken), NP_001085699.1 (frog), XP_688695.4 (zebrafish). Sequences were aligned using T-Coffee[104], with conserved and similar residues highlighted using Boxshade (https://embnet.vital-it.ch/software/BOX_form.html).

**Peptide pulldowns**. Biotinylated peptides (GenScript) were bound to streptavidin-coupled Dynabeads M-280 (Thermo Fisher Scientific). HeLa nuclear extracts (Ipracell) were diluted 1:1 with dilution buffer (300 mM NaCl, 10 mM NaF, 0.2 mM ethylenediaminetetraacetic acid (EDTA), 0.2% Igepal CA-630, 20 mM HEPES-KOH, pH 7.4) supplemented with cOmplete protease inhibitor cocktail (Roche), and cleared by centrifugation. Dynabead-conjugated peptides were incubated with clarified extracts with end-to-end mixing at 4 °C for 2 h. Beads were washed with peptide pulldown buffer (150 mM NaCl, 50 mM KCl, 5 mM NaF, 0.2 mM EDTA, 0.1% Igepal CA-630, 10% glycerol, 20 mM HEPES-KOH, pH 7.4) supplemented with cOmplete protease inhibitor cocktail, before elution in 2× SDS sample buffer (4% SDS, 20% glycerol, 50 mM TCEP, 0.002% bromophenol blue, 125 mM Tris-HCl, pH 6.8) for mass spectrometry or SDS-PAGE. Peptides were biotinylated on the N-terminus, with the following sequences: motif 1 (SGSG-KKLEFSSSPDSLSTINDWDDMDDFDTS), motif 2 (SGSG-DYDTDFVPPSPEEII and SGSG-DYDTDFVPP-[pSer]-PEEII), motif 3 (SGSG-DIDNF-DIDDFDDDDDWE), motif 4 (SGSG-ANSKLGIMAPPKPINRPFLKPSYAFS), motif 5 (SGSG-LGPSDEELLASLDENDELTAN), and motif 6 (SGSG-DGELDDFSLEEALLLEETVQKE).

For pulldowns with UV-crosslinking, biotinylated peptides containing BPA (Genosphere Biotechnologies) were incubated with clarified HeLa nuclear extracts as above. Where indicated, samples were exposed to light from a Stratalinker UV Crosslinker 2400 (Stratagene) for 1 h on ice, followed by washing with peptide pulldown buffer at 4 °C or denaturing wash buffer (137 mM NaCl, 3 mM KCl, 0.1% Tween 20, 1% SDS, 25 mM Tris-HCl, pH 7.5) at room temperature. Samples were eluted in 2× SDS sample buffer for SDS-PAGE. BPA-modified peptides were biotinylated on the N-terminus, with the following sequences: motif 1 (SGSG-KKLEFSSSPDSLSTIND-[BPA]-DDMDDFDTS), motif 3 (SGSG-DIDNFDIDD-[BPA]-DDDDDWE) and motif 5 (SGSG-LGPSDEELLAS-[BPA]-DENDELTAN).

For pulldowns with in vitro translated heterotrimeric RPA (incorporating either RPA1^{WT} or RPA1^{R41E/Y42F}), RPA was first transcribed and translated using the TNT Quick Coupled Transcription/Translation System (Promega) according to the manufacturer's instructions. In vitro translated complexes were diluted in modified (50 mM NaCl instead of 150 mM) peptide pulldown buffer supplemented with 1 mM MgCl₂, cOmplete protease inhibitor cocktail and 25 U/mL Benzonase (Novagen). After nuclease digestion, NaCl and EDTA concentrations were adjusted to 200 mM and 2 mM, respectively, and samples were cleared by centrifugation before being used for pulldowns with biotinylated peptides as above.

**Mass spectrometry**. Peptide pulldown eluates in 2× SDS sample buffer were subjected to two rounds of chloroform-methanol precipitation[105], followed by in-solution digestion with trypsin[106]. Desalted peptides were injected into a LC-MS/MS platform consisting of a Thermo Scientific UltiMate 3000 UHPLC and Q Exactive mass spectrometer. Samples were separated with 250 nl/min flow on a 50-cm EASY-Spray column with a gradient of 2–35% acetonitrile in 5% dimethyl sulfoxide (DMSO) and 0.1% formic acid. MS1 data were acquired with a resolution of 70,000 and an AGC target of 3E6. Selected precursors were fragmented with 28% nor-malized collision energy and fragment masses acquired (MS2) with a resolution of 17,500 and an AGC target of 1E5. Selected precursors were excluded for 27 s. Mass spectrometry data were converted into mgf files by msConvert[107], and analyzed with Mascot 2.5 (Matrix Science). The *Homo sapiens* Swissprot/TrEMBL database (retrieved 15/10/2014) was used for protein identification. Precursor mass tolerance was set to 10 ppm and fragment tolerance to 0.05 Da. Trypsin was selected as enzyme (one missed cleavage), carbamidomethyl (C) as fixed modification, and oxidation (M) and deamidation (NQ) as variable modification. Keratins and peptide identifications with an ion score of <20 were dismissed, and protein identification confidence was set to 1% false-discovery rate at identity or homology threshold.

**SDS-PAGE and western blotting**. SDS-PAGE and western blotting were per-formed using 7% Tris-Bicine gels with the SE400 and TE42 systems from Hoefer, or Bolt 4–12% Bis-Tris Plus gels from Thermo Fisher Scientific. The following anti-bodies were used at the indicated dilutions: BLM (A300-110A, Bethyl Laboratories, 1/2000), FLAG (F1804, Sigma-Aldrich, 1/2000), GFP (11814460001, Roche, 1/5000), MRE11 (PC388, Merck, 1/500), RMI1 (NB100-1720, Novus Biologicals, 1/1000), RMI2 (NBP1-89962, Novus Biologicals, 1/4000), RPA1 (NA13, Merck, 1/200), RPA2 (ab10359, Abcam, 1/10,000), RPA3 (HPA005708, Sigma-Aldrich, 1/1000), TOP3A (14525-1-1AP, Proteintech, 1/1000), TOPBP1 (A300-111A, Bethyl Laboratories, 1/2000), XPA (ab65963, Abcam, 1/2000). Uncropped and unprocessed scans of all blots can be found in the Source Data file associated with this paper.

**Cell lines, cell culture conditions, and RNA interference**. All cells were grown in humidified incubators supplied with 5% $CO_2$ and maintained at 37 °C, with regular testing to verify mycoplasma-free status using a LookOut Mycoplasma PCR Detection Kit (Sigma-Aldrich). RPE-1 FRT/TR cells[108] were purchased from Ximbio, and cultured in Dulbecco's modified Eagle's medium (DMEM; Thermo Fisher Scientific) supplemented with 10% fetal calf serum (FCS; Thermo Fisher Scientific) and 100 µ/ml penicillin/100 µg/ml streptomycin (Lonza). 293FT cells were obtained from Thermo Fisher Scientific and cultured in DMEM supplemented with 10% FCS, 2 mM glutamine (Lonza), 1% MEM non-essential amino acids (Thermo Fisher Scientific) and 500 µg/ml Geneticin (G418; Thermo Fisher Scientific). siRNAs were transfected using Lipofectamine RNAiMAX (Thermo Fisher Scientific) according to the manufacturer's instructions, with the following sequences: siCtrl (targeting firefly luciferase), 5′-CGUACGCGGAAUACUUCGA-3′; siRMI1, 5′-AGCCUUCACGAAUGUUGAU-3′.

**Plasmids and cloning**. All plasmids were maxiprepped using GenElute HP Endotoxin-Free Plasmid Maxiprep Kits (Sigma-Aldrich) according to the manufacturer's instructions, and verified by Sanger sequencing (Source BioScience). Plasmids expressing GFP-RMI1 (note the abbreviation "GFP" is used to include the various different *A. victoria* green fluorescent protein variants used in this study with similar excitation and emission wavelengths, including EGFP and Clover variants[109]) were constructed by subcloning human RMI1 cDNA (a gift from Andrew Deans) into mClover2-C1 (a gift from Michael Davidson; Addgene plasmid # 54577). Deletions of motifs 5 (residues 240–260) and 6 (residues 308–329) to generate plasmids expressing GFP-RMI1$^{\Delta 5}$ and GFP-RMI1$^{\Delta 6}$, respectively, were carried out by site-directed mutagenesis using QuikChange (Agilent Technologies).

Plasmids expressing GFP-tagged or FLAG-tagged BLM[25] were mutagenized to produce constructs with deletions in the TR domain (Δ4–37), motif 1 (Δ137–163), motif 2 (Δ292–309), and/or motif 3 (Δ552–568). Vectors expressing WT or R41E/Y42F RPA1 were a gift from Marc Wold[40]. Primers used in this study are listed in Supplementary Data 9.

**Immunoprecipitations**. Plasmids were transfected into 293FT cells using Lipofectamine 2000 (Thermo Fisher Scientific) according to the manufacturer's instructions. For preparation of lysates for IPs, cells were washed in phosphate-buffered saline (PBS), and lysed in IP buffer (100 mM NaCl, 0.2% Igepal CA-630, 1 mM $MgCl_2$, 10% glycerol, 5 mM NaF, 50 mM Tris-HCl, pH 7.5), supplemented with cOmplete EDTA-free protease inhibitor cocktail and 25 U/mL Benzonase. After nuclease digestion, NaCl and EDTA concentrations were adjusted to 200 mM and 2 mM, respectively, and lysates were cleared by centrifugation. Lysates were then incubated with 15 µl of GFP-Trap magnetic agarose beads (ChromoTek) for 2 h with end-to-end mixing at 4 °C. Immunoglobulin–antigen complexes were washed five times with IP buffer before elution in 2× SDS sample buffer for SDS-PAGE.

**Generation of BLM knockout RPE-1 cells**. Small guide RNAs (sgRNAs) were designed to target exon 4 of *BLM* using CRISPR Design Tool (https://crispr.mit.edu), targeting the following sequences: sgRNA-1 (anti-sense), CCACCTTCTC-CAGAAGAAATTAT; sgRNA-2 (sense), CTTCCTCTTCAAAATGCCTTAGG. sgRNA oligonucleotides (Sigma-Aldrich) were cloned into AIO-GFP (a gift from Steve Jackson[110]), to produce a vector expressing Cas9$^{D10A}$ nickase, the two sgRNAs, and GFP for sorting successfully transfected cells by FACS. Plasmids were transfected into RPE-1 FRT/TR cells using Lipofectamine 3000 (Thermo Fisher Scientific), according to the manufacturer's instructions. Cells were allowed to recover for 3 days before sorting GFP-positive cells. Colonies were grown from single cells and screened for BLM expression by western blotting.

**SCE assays**. RPE-1 cells were treated with 10 µM BrdU for two cell cycles, and blocked in mitosis by addition of 750 nM colchicine for 2 h before harvesting. Cells were resuspended in 75 mM KCl and incubated at 37 °C for 30 min, followed by fixation by addition of ice-cold 3:1 methanol/acetic acid solution. Following gentle inversion and centrifugation, pellets were resuspended in fixative and spun down, with this procedure being repeated twice. Pellets were then resuspended in fixative, and cells were dropped onto glass slides that had been pre-treated with 0.432% HCl for 1 h, washed six times in water, and stored in ethanol. Cells were left to dry at room temperature for up to 3 days in the dark, before staining by immersion in 15 µg/ml Hoechst diluted in 100 mM phosphate buffer (51 mM $NaH_2PO_4$, 49 mM $Na_2HPO_4$, pH 6.8) for 30 min. Cells were washed in PBS, exposed to a 365-nm UV light source for 2 h, and washed again in PBS. This was followed by incubation in 2× saline-sodium citrate buffer (300 mM NaCl, 30 mM trisodium citrate-HCl, pH 7) for 1 h at 65 °C and four washes in water. Cells were stained with Leishman's stain (Sigma-Aldrich) for 90 s and left in the dark for 30 min to dry. Cells were then washed once in water, left to dry overnight in the dark before mounting in VECTASHIELD (Vector Laboratories) with glass coverslips (Appleton Woods). Images were acquired using a ZEISS Axio Observer Z1 widefield microscope fitted with an Axiocam MRm camera and a Plan-Apochromat ×100/1.4 oil objective, running on ZEN Pro 2012 software.

**DNA-end resection assay**. Cells were incubated with 10 µM 5-ethynyl-2'-deoxyuridine (EdU) and 1 µM CPT for 1 h as indicated. Cells were harvested by trypsinization, pre-extracted in 0.2% Triton X-100 in PBS for 10 min on ice, and fixed using BD Cytofix/Cytoperm solution (BD Biosciences) according to the manufacturer's instructions. Cells were labeled with RPA2 antibodies (ab2175, Abcam, 1/200) diluted in BD Perm/Wash buffer for 1 h at room temperature, followed by incubation with Alexa Fluor 647 goat anti-mouse IgG1 secondary antibodies (A21240, Thermo Fisher Scientific, 1/200) diluted in BD Perm/Wash buffer for 1 h at room temperature in the dark. EdU was labeled with Alexa Fluor 555 using a Click-iT EdU Cell Proliferation Kit for Imaging (Thermo Fisher Scientific) according to the manufacturer's instructions. After washing in BD Perm/Wash buffer, cells were resuspended in PBS containing 250 µg/ml RNase A (Sigma-Aldrich), 0.02% sodium azide and 2 µg/ml DAPI for 30 min at 37 °C in the dark. Samples were resuspended in BD Perm/Wash buffer for analysis. Cell cycle profiles were generated using DAPI content and EdU intensity. DNA-end resection was quantified by subtracting the RPA2 intensity of untreated S phase cells from the RPA2 intensity of CPT-treated S phase cells, and results normalized to WT RPE-1 cells treated with siCtrl. See Supplementary Fig. 6 for further details of gating strategy.

**Clonogenic survival assays**. Cells were plated at low densities and treated with the indicated doses of HU for 24 h the following day. Cells were washed three times in PBS before addition of fresh medium, and left for up to 14 days to allow colonies to develop. Colonies were washed in PBS, stained in 0.1% Coomassie Brilliant Blue R 250, 7% acetic acid, 50% methanol at room temperature for 30 min, and washed in water before counting.

**DNA fiber analyses**. Cells were treated with 25 µM 5-iodo-2'-deoxyuridine (IdU) for 30 min, washed three times in PBS, before addition of 2 mM HU for 2 h. Cells were then washed three times in PBS before addition of 250 µM 5-chloro-2'-deoxyuridine (CldU) for 30 min. Cells were washed in PBS and harvested using Accutase (Thermo Fisher Scientific), before being embedded in low melting point agarose (Bio-Rad). Agarose plugs were solidified at 4 °C for 30 min, then incubated overnight at 50 °C in 20 mM NaCl, 100 mM EDTA, 0.5% SDS, 10 mM Tris-HCl pH 8 solution containing 1 mg/ml proteinase K (Sigma-Aldrich). Digested proteins were removed by two washes in 10 mM Tris-HCl, 1 mM EDTA (pH 8). Plugs were used immediately or stored at 4 °C in 500 µM EDTA (pH 8). Plugs were melted at 67 °C for 30 min, and cooled to 42 °C before addition of β-Agarase (New England Biolabs) according to the manufacturer's instructions. Plugs were left to digest overnight and the following day, MES buffer (pH 5.4) was added. DNA solution was heated to 65 °C for 30 min, and samples were left to cool to room temperature before transfer to the Teflon reservoir of a FiberComb Molecular Combing System (Genomic Vision). Silanized coverslips were dipped into the reservoir for 15 min and pulled out at a constant speed of 300 µm/s. Coverslips were baked overnight at 60 °C, and then incubated in 12 M HCl for 45 min with gentle shaking to denature the DNA. Coverslips were washed three times in PBS, before incubation in blocking buffer (1% bovine serum albumin, 0.2% Tween 20 in PBS) for 15 min. Samples were then incubated in the following primary antibodies diluted in blocking buffer for 1 h: anti-BrdU (347580, BD Biosciences, 1/25) to recognize IdU, and anti-BrdU (ab6326, Abcam, 1/400) to recognize CldU. Samples were washed three times in 0.2% Tween 20 in PBS, before incubation in the following secondary antibodies from Thermo Fisher Scientific diluted 1/500 in blocking buffer for 45 min: goat anti-rat IgG Alexa Fluor 488 (A11006) and donkey anti-mouse IgG Alexa Fluor 568 (A10037). Samples were washed three times in 0.2% Tween 20 in PBS and once in PBS alone, before mounting in ProLong Gold Antifade Mountant (Thermo Fisher Scientific). DNA fibers were analyzed using Fiji[111].

**Lentiviral transductions**. Generation of lentiviral vectors for complementation of $BLM^{-/-}$ RPE-1 cells with GFP-BLM and FLAG-RMI1 was done as follows. First, DNA encoding mClover3 followed by a multiple cloning site was synthesized (Thermo Fisher Scientific) and cloned into pcDNA5/FRT/TO-neo (a gift from Jon Pines; Addgene plasmid # 41000) to generate pcDNA5/FRT/TO-neo-mClover3. cDNAs encoding WT BLM or BLM lacking motifs 1 and 3 were subcloned into this vector to generate pcDNA5/FRT/TO-neo-mClover3-BLM$^{WT}$ and pcDNA5/FRT/TO-neo-mClover3-BLM$^{\Delta RPA}$. DNA encoding P2A-myc-3xFLAG-RMI1 was synthesized with silent mutations to render it siRMI1-resistant (Thermo Fisher Scientific) and cloned into pcDNA5/FRT/TO-neo-mClover3-BLM$^{WT}$ to generate pcDNA5/FRT/TO-neo-mClover3-BLM$^{WT}$-P2A-myc-3xFLAG-RMI1$^{WT}$, and into pcDNA5/FRT/TO-neo-mClover3-BLM$^{\Delta RPA}$ to generate pcDNA5/FRT/TO-neo-mClover3-BLM$^{\Delta RPA}$-P2A-myc-3xFLAG-RMI1$^{WT}$. The latter construct was mutagenized to delete motif 5 in RMI1 (residues 240–260) and generate pcDNA5/FRT/TO-neo-mClover3-BLM$^{\Delta RPA}$-P2A-myc-3xFLAG-RMI1$^{\Delta RPA}$. The WT and ΔRPA mClover3-BLM-P2A-myc-3xFLAG-RMI1 sequences were amplified by PCR, cloned into pDONR221 using Gateway Technology (Thermo Fisher Scientific), and then into pLenti PGK Neo DEST (a gift from Eric Campeau and Paul Kaufman[112]; Addgene plasmid # 19067) according to the manufacturer's instructions.

To assemble lentiviruses, 293FT cells were transfected with pLenti PGK Neo DEST plasmids encoding WT or mutant GFP-BLM and FLAG-RMI1 proteins or

GFP alone, along with pHDM-tat1b, pHDM-G, pRC/CMV-rev1b, and pHDM-Hgpm2 lentiviral assembly plasmids (all gifts from Ross Chapman[113]). Viral supernatants were harvested at 24 and 48 h, pooled, passed through a 0.45-µm filter, and stored at −80 °C.

For lentiviral infections, RPE-1 cells were incubated in viral media for 48 h, before selection in fresh medium containing 500 µg/ml G418 for up to 7 days. GFP-positive cells were sorted by FACS to generate stable lines for further study, followed by verification by western blotting.

**Immunofluorescence and confocal microscopy.** For analyses of BLM foci, RAD51 foci or UFBs, cells were grown on 18 × 18 mm #1.5H glass coverslips (Marienfeld Superior). For laser microirradiation, cells were grown in 35-mm #1.5H glass bottom dishes (Thistle Scientific). Cells were washed once in PBS before fixing for 10 min in 4% paraformaldehyde (PFA) in PBS at room temperature. When imaging UFBs, cells were instead fixed in 4% PFA in 250 mM HEPES (pH 7.0-7.6) and 0.1 % Triton X-100 for 20 min at 4 °C. PBS was added simultaneously during PFA removal to prevent samples from drying due to PFA evaporation. To visualize BLM recruitment, a mild pre-extraction step (ice-cold 0.2% Triton X-100 in PBS for 1 min followed by a wash in PBS) was carried out between the first PBS wash and the fixation step. After fixation, cells were permeabilized in 0.2% Triton X-100 in PBS for 5 min (or 15 min when imaging UFBs), and then washed once more in PBS before incubation with primary antibodies diluted in antibody buffer (DMEM supplemented with 10% FCS and 0.05% sodium azide; filtered) for 1 h. Where indicated, EdU was labeled with Alexa Fluor 647 using a Click-iT EdU Cell Proliferation Kit for Imaging (Thermo Fisher Scientific) according to the manufacturer's instructions prior to antibody incubations. The following primary antibodies were used at the indicated concentrations: Cyclin A (611268, BD Biosciences, 1/200), GFP (PABG1, ChromoTek, 1/500), γH2AX (05-636, Merck, 1/500), γH2AX (2577, Cell Signaling Technology, 1/500), RAD51 (70-002, BioAcademia, 1/1000), RIF1 (sc-515573, Santa Cruz Biotechnology, 1/50), RPA2 (ab2175, Abcam, 1/300). Cells were washed three times in 0.2% Tween 20 in PBS, before incubation with secondary antibodies diluted in antibody buffer containing 0.5 µg/mL DAPI in the dark for 30 min, followed by five washes in 0.2% Tween 20 in PBS. The following secondary antibodies were from Thermo Fisher Scientific and diluted 1/1000: goat anti-rabbit IgG Alexa Fluor 488 (A11034), donkey anti-mouse IgG Alexa Fluor 568 (A10037), donkey anti-rabbit IgG Alexa Fluor 568 (A10042) and goat anti-mouse IgG1 Alexa Fluor 647 (A21240). Cells in glass bottom dishes were imaged directly in PBS, whereas cells on coverslips were washed once in water and mounted in VECTASHIELD (unless imaging samples with Alexa Fluor 647, in which case SlowFade Diamond Antifade Mountant (Thermo Fisher Scientific) was used to avoid fluorescence quenching[114]).

All images were acquired using an LSM 880 Airyscan inverted microscope (ZEISS) equipped with a ×63/1.4 NA Plan-Apochromat objective and an Airyscan 32-pinhole detector unit. Foci were quantified using Fiji. For Airyscan imaging, raw data were processed using Airyscan processing with Wiener Filter strength 6 using ZEN Black software version 2.1, yielding 16-bit images with approximately 180-nm lateral resolution. Display of images was adjusted for intensity for optimal display of structures of interest, and false-colored in magenta or cyan where indicated.

For laser microirradiation, cells were grown for 24 h in the presence of 10 µM BrdU, before being exposed to the 405-nm diode laser from the same microscope used for imaging. Laser energy output was determined by biological calibration.

**Statistics and reproducibility.** All experiments in the paper were performed independently at least twice, with similar results.

**Reporting summary.** Further information on research design is available in the Nature Research Reporting Summary linked to this article.

## Data availability
Datasets generated for this study are available from the corresponding author upon reasonable request. Proteomics data have been deposited to the ProteomeXchange Consortium via the PRIDE partner repository[115] with the data set identifier ["PXD018322" http://www.ebi.ac.uk/pride/archive/projects/PXD018322]. Source data are provided with this paper.

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

# ARTICLE

93. Lecointe, F. et al. Anticipating chromosomal replication fork arrest: SSB targets repair DNA helicases to active forks. *EMBO J.* **26**, 4239–4251 (2007).

94. Higgs, M. R. et al. BOD1L is required to suppress deleterious resection of stressed replication forks. *Mol. Cell* **59**, 462–477 (2015).

95. Bugreev, D. V., Yu, X., Egelman, E. H. & Mazin, A. V. Novel pro- and anti-recombination activities of the Bloom's syndrome helicase. *Genes Dev.* **21**, 3085–3094 (2007).

96. Crickard, J. B. et al. The RecQ helicase Sgs1 drives ATP-dependent disruption of Rad51 filaments. *Nucleic Acids Res.* **47**, 4694–4706 (2019).

97. Hu, Y. et al. RECQL5/Recql5 helicase regulates homologous recombination and suppresses tumor formation via disruption of Rad51 presynaptic filaments. *Genes Dev.* **21**, 3073–3084 (2007).

98. van Brabant, A. J. et al. Binding and melting of D-loops by the Bloom syndrome helicase. *Biochemistry* **39**, 14617–14625 (2000).

99. Bachrati, C. Z., Borts, R. H. & Hickson, I. D. Mobile D-loops are a preferred substrate for the Bloom's syndrome helicase. *Nucleic Acids Res.* **34**, 2269–2279 (2006).

100. Fasching, C. L., Cejka, P., Kowalczykowski, S. C. & Heyer, W.-D. Top3-Rmi1 dissolve Rad51-mediated D loops by a topoisomerase-based mechanism. *Mol. Cell* **57**, 595–606 (2015).

101. Piazza, A. et al. Dynamic processing of displacement loops during recombinational DNA repair. *Mol. Cell* **73**, 1255–1266 (2019). e4.

102. LaRocque, J. R. et al. Interhomolog recombination and loss of heterozygosity in wild-type and Bloom syndrome helicase (BLM)-deficient mammalian cells. *Proc. Natl Acad. Sci. USA* **108**, 11971–11976 (2011).

103. Blackford, A. N. & Jackson, S. P. ATM, ATR, and DNA-PK: the trinity at the heart of the DNA daage response. *Mol. Cell* **66**, 801–817 (2017).

104. Notredame, C., Higgins, D. G. & Heringa, J. T-Coffee: a novel method for fast and accurate multiple sequence alignment. *J. Mol. Biol.* **302**, 205–217 (2000).

105. Wessel, D. & Flügge, U. I. A method for the quantitative recovery of protein in dilute solution in the presence of detergents and lipids. *Anal. Biochem.* **138**, 141–143 (1984).

106. Fischer, R. & Kessler, B. M. Gel-aided sample preparation (GASP)–a simplified method for gel-assisted proteomic sample generation from protein extracts and intact cells. *Proteomics* **15**, 1224–1229 (2015).

107. Chambers, M. C. et al. A cross-platform toolkit for mass spectrometry and proteomics. *Nat. Biotechnol.* **30**, 918–920 (2012).

108. Mansfeld, J., Collin, P., Collins, M. O., Choudhary, J. S. & Pines, J. APC15 drives the turnover of MCC-CDC20 to make the spindle assembly checkpoint responsive to kinetochore attachment. *Nat. Cell Biol.* **13**, 1234–1243 (2011).

109. Bajar, B. T. et al. Improving brightness and photostability of green and red fluorescent proteins for live cell imaging and FRET reporting. *Sci. Rep.* **6**, 20889 (2016).

110. Chiang, T.-W. W., le Sage, C., Larrieu, D., Demir, M. & Jackson, S. P. CRISPR-Cas9(D10A) nickase-based genotypic and phenotypic screening to enhance genome editing. *Sci. Rep.* **6**, 24356 (2016).

111. Schindelin, J. et al. Fiji: an open-source platform for biological-image analysis. *Nat. Methods* **9**, 676–682 (2012).

112. Campeau, E. et al. A versatile viral system for expression and depletion of proteins in mammalian cells. *PLoS ONE* **4**, e6529 (2009).

113. Becker, J. R. et al. The ASCIZ-DYNLL1 axis promotes 53BP1-dependent non-homologous end joining and PARP inhibitor sensitivity. *Nat. Commun.* **9**, 5406 (2018).

114. Arsić, A., Stajković, N., Spiegel, R. & Nikić-Spiegel, I. Effect of Vectashield-induced fluorescence quenching on conventional and super-resolution microscopy. *Sci. Rep.* **10**, 6441 (2020).

115. Perez-Riverol, Y. et al. The PRIDE database and related tools and resources in 2019: improving support for quantification data. *Nucleic Acids Res.* **47**, D442–D450 (2019).

116. Ishihama, Y. et al. Exponentially modified protein abundance index (emPAI) for estimation of absolute protein amount in proteomics by the number of sequenced peptides per protein. *Mol. Cell Proteom.* **4**, 1265–1272 (2005).

## Acknowledgements

We thank Ross Chapman, Michael Davidson, Andrew Deans, Steve Jackson, Jon Pines, Marc Wold, Eric Campeau, and Paul Kaufman for providing valuable reagents. We are grateful to David Clynes, Ester Hammond, and Steve Jackson for critical reading of the manuscript and helpful discussions. We thank Jill Brown, Rosa Camarillo, and Pablo Huertas for help setting up DNA combing, Nausica Arnoult and Alan Saghatelian for providing BPA peptide pulldown advice, Rebecca Konietzny for help with mass spectrometry, and Jill Brown and Wojciech Niedzwiedz for assistance with SCE assays. We acknowledge Kevin Clark, Sally-Ann Clark, Paul Sopp, and Craig Waugh in the MRC Weatherall Institute of Molecular Medicine (WIMM) Flow Cytometry Facility for providing cell-sorting services and technical expertize; Dominic Waithe, Silvia Galliani, Jana Koth, and Christoffer Lagerholm in the Wolfson Imaging Centre for help with microscopy; and Ryan Beveridge in the MRC WIMM Virus Screening Facility for help in generation of lentiviral preparations. These facilities are supported by the MRC Molecular Hematology Unit (MC_UU_12009), the MRC Human Immunology Unit (MC_UU_12010), the Wolfson Foundation (grant 18272), the Wellcome Trust (Micron 107457/Z/15Z), the National Institute for Health Research Oxford Biomedical Research Centre (IS-BRC-1215-20008), the Cancer Research UK (CRUK) Oxford Centre, the Kay Kendall Leukemia Fund (KKL1057), the John Fell Fund (131/030 and 101/517), the EPA fund (CF182 and CF170) and the MRC WIMM Strategic Alliance (G0902418 and MC_UU_12025). The Blackford lab is supported by a CRUK Career Development Fellowship (C29215/A20772) and an Against Breast Cancer/Oriel College Research Fellowship to A.N.B., and an MRC WIMM DPhil Prize Studentship (MR/N013468/1) to A.-M.K.S. K.T. is supported by the Tokyo Tech Academy for Co-creative Education of Environment and Energy Science, the Tokyo Tech Academy for Leadership, and a Grant-in-Aid Fellowship from the Japan Society for the Promotion of Science (JP20J13601). J.S. is supported by a Tetelman Fellowship for International Research in the Sciences from Yale College.

## Author contributions

The project was conceived and supervised by A.N.B. Experiments were performed by A.-M.K.S., S.E.J., K.T., Z.B., J.S., and A.N.B. Reagents were generated by S.E.J., K.T., C.A.M., and A.N.B. Mass spectrometry data were contributed by I.V., R.F., and B.M.K. The paper was written by A.N.B. with contributions from A.-M.K.S., S.E.J., and K.T.

## Competing interests

The authors declare no competing interests.
