## [Peer Review File · Nature Communications]

REVIEWER COMMENTS

Reviewer #1 (Remarks to the Author):

Summary: In this work, the authors have investigated the functional importance of protein interactions of the single-stranded DNA binding protein RPA with BLM helicase and the interacting factor RMI1, which together with RMI2 and TOP3A constitute the BTR complex responsible for double HJ dissolution that contributes to the suppression of elevated sister chromatid exchange in the chromosomal instability and cancer-prone disorder Bloom's syndrome (BS). Protein interaction site mapping and cell-based assays are used to build a model in which RPA's interaction with BLM/RMI1 is important for the replication stress response but surprisingly not to suppress SCEs or DNA end-resection. It is postulated by the authors that the separation-of-function mutants provide new insight to the role of BLM in the replication stress response and will be valuable for future work to dissect the pathologies associated with BS.

Critical Comments:

The mapping results build upon and extend previously published work. The functional cell-based assays which assess that impact of site-specific mutant versions provide some insight to the apparent "separation-of-function", but a true mechanistic insight to BLM's role in the response to replication stress and the importance of the physical interactions of RPA70 to BLM or RMI1 proteins is not addressed.

Previously, Doherty et al had reported that the N-terminus of BLM is required for RPA binding (ref 66). Brosh et al. previously reported that BLM interacts with the RPA70 subunit (ref 17). While the former result is mentioned on line 303 of the Discussion, and the latter is omitted but the reference cited for the following statement "recombinant BLM suggested to bind RPA in vitro", both findings should be clearly stated in the Introduction as they strongly relevance to the current study. In addition, Doherty et al had presented evidence that a catalytically active BLM helicase domain fragment that lacks the physical binding site for RPA fails to be stimulated to unwind long DNA duplexes in vitro. This result would suggest that disruption of RPA binding to BLM would impair BLM-catalyzed unwinding of long DNA duplexes. While the precise role of BLM (and its interaction/significance with RPA) in response to DNA replication stress was not investigated in this study, the authors might more carefully consider their conclusion made on line 335 regarding a mechanistic role or importance of RPA binding to stably recruit the BTR complex to ssDNA sites.

Figure 1A should have amino acid boundary sites for the various motifs and domains. This would greatly aid the reader interested in more precisely ascertaining the locations of such motifs and domains in the linear sequence (e.g., motif 1, 2, 3, and 4 in BLM). The same applies to RMI1.

Line 114: "incubated in" should be "incubated with"

Figure 3d: GFP-BLM accumulation at laser-induced DNA damage to assess importance of RPA binding. This experiment requires a time course analysis to make sure that the results are unequivocal in the conclusions made by the authors.

Neither in the model presented in Figure 5b nor the text is it mentioned the work from the Bartek lab and collaborators that RPA pool depletion can occur during replication stress, and the consequences of this. This work has implications for the current study and should be considered.

Lines 266-269: Authors state: "RPA binding does not recruit the BTR-complex to DNA double-strand breaks to stimulate end-resection" The assay used is indirectly measuring strand resection (CPT-induced RPA intensity as a marker of ssDNA generated, not BTR recruitment).

Reviewer #2 (Remarks to the Author):

BLM is known have important functions in HJ dissolution, DSB resection, UFB resolution, and restart of stalled replication forks. Although BLM and RMI1 were shown to bind RPA, the functional significance of these bindings was not known. In this study, the authors carefully mapped the RPA binding motifs in the BTR complex. They identified two binding motifs in BLM and one in RMI1. The BTR complex lacking these RPA binding motifs cannot bind RPA and localize to laser induced DNA damage stripes in cells. Interestingly, this BTR mutant is not defective in SEC and DSB resection, but is defective for the restart of stalled replication forks after HU treatment. The results of this study help explain how BLM is regulated by DNA damage in cells, and they also suggest that the functions of BTR in fork restart can be separated from other functions. Overall, the data in this study are of high quality, and the conclusions are largely convincing. However, as discussed below, there are still a number of questions about the model that need to be addressed or clarified. A carefully revised version of this manuscript could be suitable for publication in Nature Communications.

1. In Fig. 2A, why didn't GFP-RMI1d5 pull down some RPA indirectly through BLM?
2. On page 8, the authors raised the possibility that the BTR complex may bind three RPA complexes simultaneously. Can the authors provide further evidence to support this point using BTR mutants rather than just the synthetic peptides?
3. Did BLM KO affect the cell cycle? The effects on resection could be indirect.
4. In Fig. S3e, what is the evidence that the RAD51 foci are formed at replication forks? Are these stalled forks or collapsed forks?
5. The data in Fig. 3d seem to suggest that the RPA binding of BLM/RMI1 is required for the localization of BTR to DSBs. However, in Fig. 4b, the authors concluded that BTR is not recruited by RPA to DSBs. This is quite confusing. How can they distinguish DSBs and stalled forks in Fig. 3d? Many papers have used the same condition to study the DSB response. Could BTR respond differently to laser and CPT induced DSBs?
6. The data in Fig. 4c and 4d are clear, but are these results specific to HU? CPT also induces collapsing of replication forks. Is the BLM/RMI RPA binding mutant sensitive to CPT too?
7. A more general question on the model: RPA-coated ssDNA is generated by DSB resection, so BTR should be recruited by RPA after resection. Also, dHJ structures don't contain ssDNA, ssDNA is needed to initiate HR and form dHJs. It is puzzling why the interaction between BTR and RPA is not required for DSB repair and SCE. The author should go beyond DSB resection and SCE, and test the BLM/RMI1 mutant in functional HR assays.
8. What about the function of BLM in resolving UFBs? Can the authors test that too?

RESPONSE TO REVIEWERS

We thank both reviewers for their positive and thoughtful comments and suggestions as to how to improve our manuscript, which we have addressed in detail below.

Reviewer #1 (Remarks to the Author):

Summary:

In this work, the authors have investigated the functional importance of protein interactions of the single-stranded DNA binding protein RPA with BLM helicase and the interacting factor RMI1, which together with RMI2 and TOP3A constitute the BTR complex responsible for double HJ dissolution that contributes to the suppression of elevated sister chromatid exchange in the chromosomal instability and cancer-prone disorder Bloom's syndrome (BS). Protein interaction site mapping and cell-based assays are used to build a model in which RPA's interaction with BLM/RMI1 is important for the replication stress response but surprisingly not to suppress SCEs or DNA end-resection. It is postulated by the authors that the separation-of-function mutants provide new insight to the role of BLM in the replication stress response and will be valuable for future work to dissect the pathologies associated with BS.

Critical Comments:

The mapping results build upon and extend previously published work. The functional cell-based assays which assess that impact of site-specific mutant versions provide some insight to the apparent "separation-of-function", but a true mechanistic insight to BLM's role in the response to replication stress and the importance of the physical interactions of RPA70 to BLM or RMI1 proteins is not addressed.

Previously, Doherty et al had reported that the N-terminus of BLM is required for RPA binding (ref 66). Brosh et al. previously reported that BLM interacts with the RPA70 subunit (ref 17). While the former result is mentioned on line 303 of the Discussion, and the latter is omitted but the reference cited for the following statement "recombinant BLM suggested to bind RPA in vitro", both findings should be clearly stated in the Introduction as they strongly relevance to the current study.

We agree that these studies are highly relevant to our findings, and have given them both greater prominence in the Introduction section of our revised manuscript.

In addition, Doherty et al had presented evidence that a catalytically active BLM helicase domain fragment that lacks the physical binding site for RPA fails to be stimulated to unwind long DNA duplexes in vitro. This result would suggest that disruption of RPA binding to BLM would impair BLM-catalyzed unwinding of long DNA duplexes. While the precise role of BLM (and its interaction/significance with RPA) in response to DNA replication stress was not investigated in this study, the authors might more carefully consider their conclusion made on line 335 regarding a mechanistic role or importance of RPA binding to stably recruit the BTR complex to ssDNA sites.

We agree that it is possible that RPA binding to BLM might stimulate its ability to unwind certain DNA structures. That is why we originally said that "a major role" rather than "the major role" of RPA-binding is to stably recruit the BTR complex to ssDNA sites. But we agree we could make this more clear, so we now specifically state in this paragraph of the Discussion that RPA-binding could stimulate unwinding of DNA by BLM and quote the Doherty study. We also removed the word "mechanistically" from this paragraph.

Figure 1A should have amino acid boundary sites for the various motifs and domains. This would greatly aid the reader interested in more precisely ascertaining the locations of such

motifs and domains in the linear sequence (e.g., motif 1, 2, 3, and 4 in BLM). The same applies to RMI1.

Amino acid residue numbers have been added as suggested in our revised manuscript.

Line 114: “incubated in” should be “incubated with”

This has been changed in our revised manuscript.

Figure 3d: GFP-BLM accumulation at laser-induced DNA damage to assess importance of RPA binding. This experiment requires a time course analysis to make sure that the results are unequivocal in the conclusions made by the authors.

We agree and so carried out a time-course experiment, which shows that the RPA-binding BLM mutant does not at any point colocalize with RPA in ssDNA microfoci along the path of the laser line (see **new Fig. 3d**).

Neither in the model presented in Figure 5b nor the text is it mentioned the work from the Bartek lab and collaborators that RPA pool depletion can occur during replication stress, and the consequences of this. This work has implications for the current study and should be considered.

We are unsure exactly which paper the reviewer is referring to here. The study that most closely matches this description is one published in Cell in 2013 from Jiri Lukas' lab, on which Jiri Bartek was a contributing author (PMID: 24267891); so we now mention this in the Introduction of our revised manuscript to highlight the importance of RPA in protecting forks from catastrophic breakage.

Lines 266-269: Authors state: “RPA binding does not recruit the BTR-complex to DNA double-strand breaks to stimulate end-resection” The assay used is indirectly measuring strand resection (CPT-induced RPA intensity as a marker of ssDNA generated, not BTR recruitment).

We agree and have modified this sentence to read: “...RPA-binding is not required for the BTR complex to stimulate DNA-end resection.”

Reviewer #2 (Remarks to the Author):

BLM is known have important functions in HJ dissolution, DSB resection, UFB resolution, and restart of stalled replication forks. Although BLM and RMI1 were shown to bind RPA, the functional significance of these bindings was not known. In this study, the authors carefully mapped the RPA binding motifs in the BTR complex. They identified two binding motifs in BLM and one in RMI1. The BTR complex lacking these RPA binding motifs cannot bind RPA and localize to laser induced DNA damage stripes in cells. Interestingly, this BTR mutant is not defective in SEC and DSB resection, but is defective for the restart of stalled replication forks after HU treatment. The results of this study help explain how BLM is regulated by DNA damage in cells, and they also suggest that the functions of BTR in fork restart can be separated from other functions. Overall, the data in this study are of high quality, and the conclusions are largely convincing. However, as discussed below, there are still a number of questions about the model that need to be addressed or clarified. A carefully revised version of this manuscript could be suitable for publication in Nature Communications.

1. In Fig. 2A, why didn't GFP-RMI1d5 pull down some RPA indirectly through BLM?

It did, but this residual RPA was not visible in the exposure of the blot we included in the initial version of the manuscript because at higher exposures the RPA band in the GFP-RMI1-WT lane becomes over-saturated. However, we agree that this is important to show, so we have included both high and low exposures in **Fig. 1d** of our revised manuscript.

2. On page 8, the authors raised the possibility that the BTR complex may bind three RPA complexes simultaneously. Can the authors provide further evidence to support this point using BTR mutants rather than just the synthetic peptides?

We had already showed using BLM and RMI1 deletion mutants that loss of any of the three motifs individually in BLM or RMI1 causes a reduction in RPA-binding (see **Fig. 1d, e** in our revised manuscript), so each motif is functional and they are not redundant with each other. If they were, then loss of one or two motifs would not cause a defect in RPA-binding. We also showed that all three motifs bind to RPA1, rather RPA2 or RPA3 (**Fig. 2a** in our revised manuscript). However, we accept that it would still have been formally possible that despite their similarity to each other and to previously described motifs that all bind to the same basic binding cleft in the N-terminus of RPA1, the three motifs might be able to interact with a single RPA1 molecule via different binding sites. To test this, we carried out peptide pulldowns where we incubated each motif individually with in vitro translated RPA heterotrimer that was either WT or mutated in the basic binding cleft of RPA1 (R41E/Y42F). Results from these experiments clearly show that while each peptide can bind WT RPA, their binding is severely disrupted to the R41E/Y42F mutant RPA (see **new Fig. 2c**). Thus, we conclude that all three motifs bind the same site on RPA, meaning that the BTR complex has the binding sites that in theory enable it to bind to three separate RPA1 subunits (and therefore three discrete RPA complexes). Combined with our data using BTR mutants as mentioned above, it is reasonable to conclude that a single BTR complex can bind at least three RPA complexes. However, structural studies (e.g. using cryo-EM) of the BTR complex and RPA will be required to formally prove this in future.

3. Did BLM KO affect the cell cycle? The effects on resection could be indirect.

We agree that this can be a concern in other resection assays but in our FACS-based assay, we are able to assess resection specifically in the S phase population. This is because we introduce DSBs with CPT, which only causes DNA damage during replication, and we pulse-label cells with EdU to identify replicating cells. We are thus able to specifically analyse DNA-end resection in S-phase cells, which avoids any issues caused by potential differences in cell cycle profiles (although our BLM-deficient cells do not show altered cell cycle progression anyway).

4. In Fig. S3e, what is the evidence that the RAD51 foci are formed at replication forks?

It is well-established that RAD51 is recruited to both stalled and collapsed replication forks in response to HU treatment, regardless of DSB formation (e.g. Hanada et al. NSMB 2007; Sirbu et al., Genes Dev. 2011). However, we acknowledge that in our original submission we did not include a cell-cycle marker or a marker for stalled/collapsed replication forks to show colocalization with RAD51, along with representative images of non-HU-treated cells. In our revised manuscript, we correct this oversight. Firstly, by pulsing cells with EdU for 15 min to mark the location of replication forks prior to addition of HU, we show that RAD51 foci colocalize with EdU, indicating that RAD51 foci are formed at replication forks (**new Supplementary Fig. 3g**). These foci also stain positive for γ H2AX, which marks both stalled and collapsed forks (e.g. Petermann et al., Mol. Cell 2010; Sirbu et al., Genes Dev. 2011). We did not see significant RAD51 or γ H2AX foci formation in non-HU-treated cells, confirming that RAD51 is not recruited to ongoing replication forks in the absence of replication stress. Importantly, when we quantify the numbers of RAD51 foci in WT and BLM-deficient cells after HU, we find that there is no significant difference between them (**new Supplementary Fig. 3h**). We therefore believe that our original conclusion that the fork restart defect in BLM-deficient cells is not caused by excessive RAD51 accumulation is correct.

Are these stalled forks or collapsed forks?

Fork breakage begins several hours after HU treatments at millimolar doses (Hanada et al. NSMB 2007; Petermann et al., Mol. Cell 2010; Sirbu et al., Genes Dev. 2011), so at the 24-hr timepoint we use for this experiment, some cells will have low levels of fork collapse while

others will have entered a state of replication catastrophe (Toledo et al., Cell 2013); even in the latter cells, each diffraction-limited RAD51 focus probably contains a mixture of stalled and collapsed forks.

5. The data in Fig. 3d seem to suggest that the RPA binding of BLM/RMI1 is required for the localization of BTR to DSBs. However, in Fig. 4b, the authors concluded that BTR is not recruited by RPA to DSBs.

We did not conclude from these data that BTR is not being recruited to DSBs, just that there is a defect in the stable recruitment of BLM to ssDNA microfoci. Indeed, as we show in our **new Fig. 3d**, RPA-binding is not absolutely essential for BLM recruitment to laser-induced DNA damage because some BLM can still be detected there at early timepoints, prior to the appearance of RPA foci. We therefore hypothesise that this residual BLM recruitment is sufficient to promote DNA-end resection.

This is quite confusing. How can they distinguish DSBs and stalled forks in Fig. 3d? Many papers have used the same condition to study the DSB response. Could BTR respond differently to laser and CPT induced DSBs?

We accept that our original experiment using laser-induced DNA damage is not informative as to whether RPA-binding is required for BLM recruitment to stalled replication forks. We therefore went back and developed a fixation and pre-extraction protocol to allow us to visualise BLM foci induced by HU. As we show in **new Fig. 5c, d**, there was a substantial defect in the ability of the RPA-binding mutant BLM protein to co-localize with RPA at sites of replication stress induced by HU. As we saw in the laser line experiment, some recruitment of the RPA-binding BLM mutant still occurred, but this might be expected given that each HU-induced focus probably represents a replication factory including a mixture of both stalled and collapsed forks. We speculate that residual mutant BLM foci could represent BLM binding directly to DNA ends at collapsed forks to promote resection (which does not require RPA-binding; **Fig. 4d**); alternatively, they could represent dHJs, which BLM can also process without needing to bind RPA (**Fig. 4a**).

6. The data in Fig. 4c and 4d are clear, but are these results specific to HU? CPT also induces collapsing of replication forks. Is the BLM/RMI RPA binding mutant sensitive to CPT too?

This would be an interesting experiment, so we began by testing how sensitive our BLM^{-/-} clones were to CPT. However, in our hands BLM deficiency causes only a very mild sensitivity to CPT compared to WT cells (see right inset), in line with what has been shown previously (Gravel et al., Genes Dev. 2008). We therefore could not carry out the suggested experiment in our complemented cells, and speculate that the difference in the mechanisms whereby CPT and HU cause replication stress might explain this result.

7. A more general question on the model: RPA-coated ssDNA is generated by DSB resection, so BTR should be recruited by RPA after resection.

This is indeed the case. As our new laser line time-course shows (**new Fig. 3d**), a small amount of BLM is recruited prior to RPA microfoci formation along the path of the laser line, and this occurs regardless of the presence of the RPA-binding motifs in BLM/RMI1. We speculate that this is the pool of BLM that promotes DNA-end resection. Subsequently, WT

BLM forms bright foci that colocalize with RPA foci along the path of the laser line, but this does not occur with the mutant protein.

Also, dHJ structures don't contain ssDNA, ssDNA is needed to initiate HR and form dHJs. It is puzzling why the interaction between BTR and RPA is not required for DSB repair and SCE. The author should go beyond DSB resection and SCE, and test the BLM/RMI1 mutant in functional HR assays.

[REDACTED]

8. What about the function of BLM in resolving UFBs? Can the authors test that too?

We first tested whether our BLM mutant could localize to UFBs like WT BLM, and found that it could (**new Fig. 4b**). This was expected, given that PICH, rather than RPA, recruits BLM to UFBs (Ke et al., EMBO 2011; Hengeveld et al., Dev. Cell 2015). We then tested whether our BLM knock-out clones displayed elevated numbers of UFBs, and found that they did (**new Supplementary Fig. 3b**), as expected from the literature (Chan et al., EMBO J. 2007). Finally, we examined whether there was any difference in the number of UFBs in cells expressing either WT or RPA-binding mutant BLM/RMI1 and found that the mutant proteins could correct the increased UFBs to the same extent as the WT protein (**new Fig. 4c**). Thus, we conclude that RPA-binding of the BTR complex does not play a significant role in UFB resolution. This is consistent with previous reports showing that RPA and BLM do not colocalize on UFBs, and that RPA may play a role in excluding BLM from these structures (Sarlos et al., NSMB 2018).

REVIEWERS' COMMENTS

Reviewer #2 (Remarks to the Author):

The authors have adequately addressed my comments. I recommend acceptance of this manuscript.

We thank the reviewers for recommending acceptance of our manuscript.